# The genomic landscape of Vk*MYC myeloma highlights shared pathways of transformation between mice and humans

Francesco Maura [1,4] ✉, David G. Coffey [1,4], Caleb K. Stein[2], Esteban Braggio [2], Bachisio Ziccheddu[1], Meaghen E. Sharik [2], Megan T. Du [2], Yuliza Tafoya Alvarado [2], Chang-Xin Shi[2], Yuan Xiao Zhu[2], Erin W. Meermeier [2], Gareth J. Morgan [3], Ola Landgren [1], P. Leif Bergsagel [2,4] & Marta Chesi [2,4] ✉

Multiple myeloma (MM) is a heterogeneous disease characterized by frequent MYC translocations. Sporadic MYC activation in the germinal center of genetically engineered Vk*MYC mice is sufficient to induce plasma cell tumors in which a variety of secondary mutations are spontaneously acquired and selected over time. Analysis of 119 Vk*MYC myeloma reveals recurrent copy number alterations, structural variations, chromothripsis, driver mutations, apolipoprotein B mRNA-editing enzyme, catalytic polypeptide (APOBEC) mutational activity, and a progressive decrease in immunoglobulin transcription that inversely correlates with proliferation. Moreover, we identify frequent insertional mutagenesis by endogenous retro-elements as a murine specific mechanism to activate NF-kB and IL6 signaling pathways shared with human MM. Despite the increased genomic complexity associated with progression, advanced tumors remain dependent on *MYC*. In summary, here we credential the Vk*MYC mouse as a unique resource to explore MM genomic evolution and describe a fully annotated collection of diverse and immortalized murine MM tumors.

Multiple myeloma (MM) is a malignancy of post-germinal center plasma cells (PC) consistently preceded by asymptomatic precursor conditions: monoclonal gammopathy of undetermined significance (MGUS) and smoldering multiple myeloma (SMM)[1–4]. Leveraging whole genome sequencing (WGS) it has recently been shown that patients with stable precursor conditions are characterized by either absence or lower prevalence of myeloma-defining genomic events, including complex structural variant (SV) events (e.g., chromothripsis), APOBEC mutational activity, mutations in distinct driver genes, and copy-number variation (CNV)[5,6]. Most of these myeloma-defining genomic events are acquired over time and are often detectable several years before disease progression, reflecting a long

evolutionary history that can experimentally be accelerated in murine models.

The Vk*MYC genetically engineered mouse model of MM is based on the sporadic AID-induced activation of human MYC in a single germinal center (GC) B-cell, in a mouse strain, C57Bl/6, that spontaneously develops monoclonal gammopathy. With age, Vk*MYC mice universally develop a progressive expansion of isotype class-switched, somatically hypermutated, monoclonal PC restricted to the bone marrow (BM), highlighting a dependency on the BM microenvironment for PC growth and survival, as reported for human MM[7]. Clinical myeloma defining end organ damaging events (CRAB: renal impairment, anemia, bone disease) occur only after long latency (usually 70

[1]Division of Myeloma, University of Miami, Miami, FL, USA. [2]Department of Medicine, Division of Hematology and Medical Oncology, Mayo Clinic, Scottsdale, AZ, USA. [3]Myeloma Research Program, NYU Langone, Perlmutter Cancer Center, New York, NY, USA. [4]These authors contributed equally: Francesco Maura, David G. Coffey, P. Leif Bergsagel, Marta Chesi. ✉e-mail: fxm557@med.miami.edu; chesi.marta@mayo.edu

weeks of age), suggesting that additional mutations, beside MYC dysregulation, are required to induce full malignant transformation[7]. An accelerated disease course is noted when MM tumors harvested from aged, de novo Vk*MYC mice are transplanted into syngeneic, non-irradiated C57BL/6 wild type mice[8]. Despite early evidence from aCGH and a small series of Vk*MYC MM interrogated by scRNA that revealed a complex and heterogenous landscape associated with Vk*MYC MM progression[9,10], it is largely unclear if these genomic events recapitulate what is observed in human.

In this study, we utilized a multi-omics approach (Supplementary Fig. 1) to illustrate that the clinically relevant, immunocompetent Vk*MYC mouse model can faithfully replicate some of the key genomic events that are spontaneously acquired and selected during the progression of MM in humans[7]. Interrogating the genomic and/or transcriptomic landscape of 128 Vk*MYC MM samples representing 119 unique tumors from 41 de novo MM, 65 transplantable lines and 25 tumors capable of growing in vitro (Supplementary Fig. 1), we reveal that phenotypic similarities of the Vk*MYC mouse to human MM are driven by spontaneously acquired common genomic events, including NFkB activation, APOBEC mutational activity, driver gene mutations, aneuploidies and complex structural variants.

## Results

### Landscape of SNV in driver genes in Vk*MYC MM

Tumor DNA from Vk*MYC MM was analyzed using WGS ($n = 41$) and WES ($n = 27$), and control tail DNA using WGS ($n = 3$). This cohort included 15 aged mice that spontaneously developed MM (i.e., de novo), 33 recipient mice transplanted with serially passaged tumor cells from de novo donors (i.e., transplanted), and 20 in vitro cultured tumor cells derived from either de novo or transplanted mice (i.e., in vitro), that we propose correspond to newly diagnosed MM, relapse multiple myeloma and myeloma cell lines, respectively (Supplementary Data 1–3). Five independent Vk*MYC tumors lines were interrogated at different stages of MM progression: de novo, transplant and in vitro for Vk12598 MM; transplant and in vitro for Vk12653, Vk31159, and Vk37553 MM; de novo and transplant for Vk36040 MM. The overall Vk*MYC MM total mutational burden was 3.8/Mb (range 0.6–20). Importantly, the number of variants per Mb per sample was not significantly different between Vk*MYC WGS and WES, indicating a lack of bias in the sequencing technology for detecting coding variants (Supplementary Fig. 2A).

To identify driver genes under positive selection, the ratio of nonsynonymous to synonymous substitutions (dN/dS) was tested using the *dndscv* R[11] package, both considering all mutations and restricting to a catalogue of oncodrivers generated by combining known MM driver genes and the COSMIC census[12–14]. A total of 11 driver genes under positive selection were detected (q < 0.1; Fig. 1A, Supplementary Fig. 2B and Supplementary Data 4), six of which are related to known driver genes in MM (*Dusp2, Trp53, Nfkbia, Pim1, Tent5c/Fam46c, H1f4/Hist1h1e*). Positively selected genes currently not included in the catalogue of recurrent MM drivers included *H1f2/Hist1h1c, Pten*, and *Tbsb4x*. Additional non-synonymous SNVs were found in known human MM driver genes such as *Nras, Kras, Cyld, Sp140, Rasa2*, and *Dis3*, but at a lower frequency compared to human MM (Fig. 1A). The non-synonymous mutational burden progressively increased across the different stages of MM progression: de novo (median 18 per sample; range 3–39), transplanted (median 43 per sample; range 9–149), and in vitro (median 74 per sample; range 19–273) (Fig. 1B, Supplementary Data 5). Interestingly, the nonsynonymous mutational burden of de novo samples was similar to the one observed in stable human precursor conditions, and the one observed in transplanted samples was similar to that of human MM and progressive precursor conditions (Fig. 1B, Supplementary Data 5). *Tp53, Dusp2*, and *H1f4/Hist1h1e* mutations were significantly enriched in MM tumors growing in vitro, frequently co-occurring in the same sample (*p* < 0.05 using

Fisher's Exact test; Fig. 1C; Supplementary Fig. 2C, Supplementary Data 6). This is in line with results in human MM, with *TP53* mutations in 57% of cell lines vs 2% of NDMM, and *DUSP2* mutations in 10% of cell lines versus 5% NDMM based on an WES analysis of 69 human myeloma cell lines (https://www.keatslab.org) and the CoMMpass dataset. Additional patterns of co-occurrence between mutated driver genes were observed, such as between *Tent5c/Fam46c* and *H1f4/Hist1h1e* (*p* < 0.05 using Fisher's exact test; Supplementary Fig. 2C). Interestingly, the variant allele frequency corrected for purity (VAF) for 10 out of 11 clonal nonsynonymous mutations in *Pten* and seven out of nine in *Trp53* were greater than 75% (Supplementary Fig. 2D). Although proper characterization of the allele-specific haplotype was not possible since the mouse genome of inbred strains is enriched for homozygous germline SNPs, the fact that only three of these clonal mutations were associated with copy number loss suggests recurrent presence of copy number neutral loss of heterozygosity (CN-LOH). It is notable that individual samples have multiple coding and non-coding mutations clustered in the first 2 kb of *Dusp2* and *Pten*, with a preference for T (27/40 SNV involve T) in the latter, suggesting the involvement of a transcription linked mutational process (Supplementary Fig. 2E, F).

### Recurrent copy number alterations in Vk*MYC MM

To characterize the landscape of CNV in Vk*MYC MM, we combined our WES and WGS cohort with an additional 28 cases interrogated by aCGH, respectively (Supplementary Data 7). For CNV calls in 11 Vk*MYC MM tumors with WES and mate-pair WGS data, we selected the latter. Applying GISTSIC2.0 to 96 Vk*MYC MM we uncovered recurrent arm-level (n = 7; 4 gains and 3 deletions) and focal (n = 20; 5 gains and 15 deletions excluding the immunoglobulin loci) CNV in the Vk*MYC MM genome (Fig. 2A, B; Supplementary Fig. 3; Supplementary Data 8, 9). As we previously noted, large chromosome 5 loss was observed in 27% of cases confirming the driver role of this large deletion in the Vk*MYC MM[9,10]. Moreover, we confirmed chromosome 14 was the second largest chromosomal loss that emerged as enriched in our cohort (17%). Interestingly, chromosome 14 in mice is largely syntenic with chromosome 13q, the most frequent large chromosomal loss in human MM (Supplementary Figs. 4, 5)[9]. Large and focal gain on chromosome 3 (syntenic to human 1q) and *MCL1* were observed in 12.5%. Several known cancer and human MM driver genes were observed within focal GISTIC peaks, including homozygous deletions of *Cdkn2a/b* (7.3%), *Rb1* (7.3%), and *Pten* (4%) (Fig. 2A, B; Supplementary Data 8 and Supplementary Fig. 3). Although 11qB2 (syntenic to human 17p) was detected as recurrently loss by GISTIC, the peak included *Ncor1* but not *Trp53*. The lack of significance for *Trp53* (11qB3) CNV loss was due to the fact that all but two *Trp53* clonally mutated cases had a CN-LOH that is not counted by GISTIC as a loss. Investigating separate CNV on chromosome X, we noted that out of 45 female, 35 (78%) had lost one X chromosome, and out of 44 male, 30 (68%) had lost the Y chromosome. Loss of X and Y chromosomes has been identified by conventional karyotype in human MM, with a frequency of 33% and 20%, respectively[15,16]. We also confirmed the high prevalence of complete loss of *Kdm6a* (7%), known putative driver gene in human MM, in four male and three female[9,17–19]. Interestingly, of those five male, four also lost the *Kdm6a/Utx* homolog *Uty*. Moreover, analysis of M-spike over time identified that male had a highly statistically significant earlier onset of monoclonal gammopathy, with a difference in prevalence at 75 weeks of age of 55% versus 45% (Fig. 2E). This is in line with the difference in the relative prevalence in male versus female of 56% to 44% of MGUS in those over 50 years old in Olmsted County (3.7% vs 2.9%), and 63% to 37% of SMM in Iceland (0.67% vs 0.39%)[20,21].

Next, we investigated potential temporal patterns of CNV acquisition over time by comparing the genomic profile of Vk*MYC MM collected at different stages of progression (Supplementary Figs. 3, 6). Differences in the frequency of genomic abnormalities were found across de novo, transplant, and in vitro Vk*MYC MM (Fig. 2B,

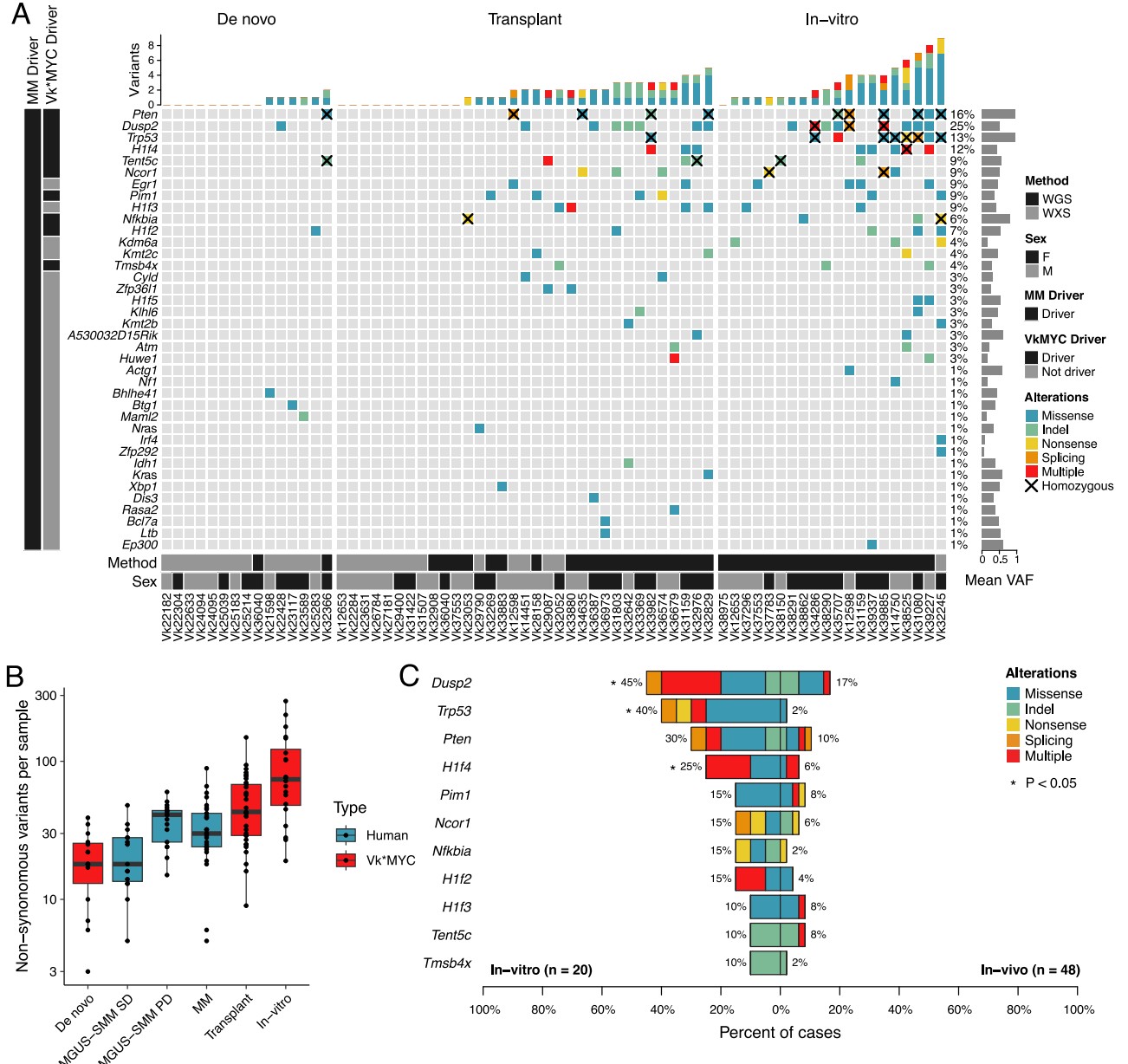

**Fig. 1 | Somatic, non-synonymous single nucleotide variants and indels in driver genes of Vk*MYC MM. A** Oncoplot of mutated driver genes detected by the dN/dS method or any nonsynonymous mutations found in a known MM driver genes. **B** Boxplot comparing nonsynonymous mutational burden of Vk*MYC MM ($n = 68$) and human MGUS/SMM with stable disease (SD = 15), MGUS/SMM with progressive disease (PD, $n = 17$), and MM ($n = 29$). Boxplots display the median, quartiles, and variability of the data. **C** Bar plot showing differentially mutated genes according to Vk*MYC MM type between in vitro and in vivo (i.e., de novo or transplant). Two-side $p$ value were estimated using Fisher exact test.

Supplementary Fig. 3; Supplementary Data 10, 11). Large gains of chromosomes 1, 3, 11, 15, and 19, and losses of chromosomes 4, 6, and 8 were more prevalent in in vitro samples than de novo, whereas gains of chromosomes 6, 7, 9, 12, and 16 occur at similar frequency across stages of MM. Focal 13qC1 deletion (*Fam172a*) as well as 3qF2.1 (*Mcl1*) amplification were also significantly more frequent among in vitro than transplant and de novo tumors ($p < 0.05$ using Fisher's exact test, Supplementary Data 11). Although deletions of the FAM172a region (5q15) do not occur in human MM, amplifications of MCL1 (1q21) are common and have been reported to be acquired with tumor progression[22]. To further expand our temporal investigations, we interrogated five individual Vk*MYC tumors with samples collected at different stages of MM progression. Signs of clonal evolution were evident at an all-chromosome level, with gains of chromosomes 1 and 17, 7, 19, and 14 with tumor progression in Vk12598, Vk12653, Vk31159

and Vk36040 MM, respectively, which could not simply be attributed to tumor purity (Supplementary Fig. 6A, B). In Vk12653 MM, we found that early passages had only one copy of chromosome 5, while late passages and in vitro line had two, suggesting a duplication of monosomic chromosome 5 with tumor progression, as recently shown using scRNA data[10]. Also in Vk12563 MM, we identified the acquisition of a *Kdm6a* biallelic deletion with bortezomib treatment (Supplementary Fig. 6C). Otherwise, we did not identify differences in focal CNA or NS-SNV of driver genes in the serial samples, suggesting the mechanisms of progression lie elsewhere.

Large chromosomal duplications (i.e., >50% of the chromosome size) and trisomies were observed in 77% of cases with evidence of multiple co-occurring trisomies in 26% cases (Fig. 2B; Supplementary Fig. 3A). To define the temporal relationship between these large clonal chromosomal duplications, we estimated the molecular time of 33

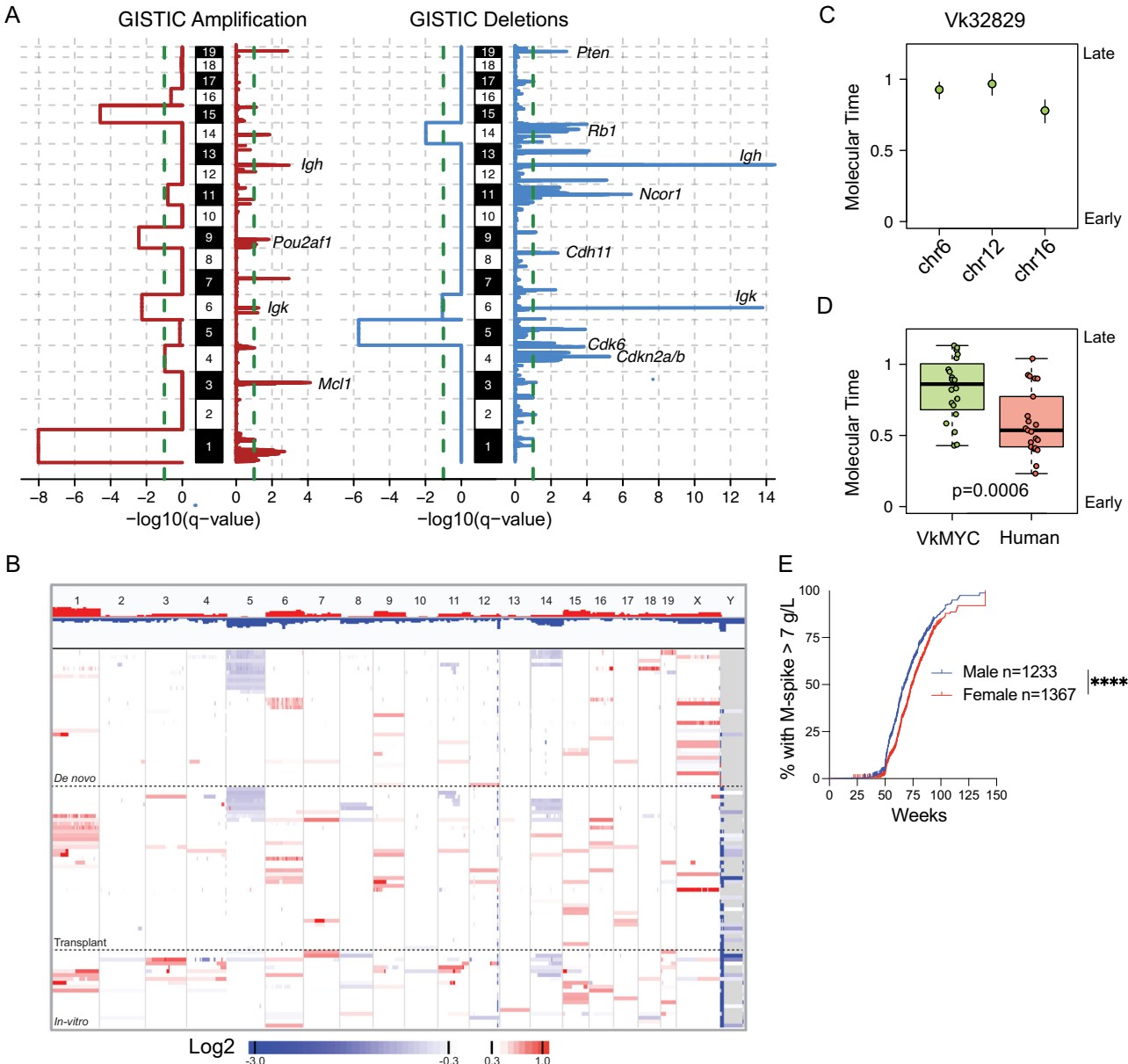

**Fig. 2 | Recurrent focal and broad copy number alterations in Vk*MYC MM.**
**A** Significant CNA GISTIC2.0 whole chromosome and focal peaks and involved genes across 96 Vk*MYC MM mice. **B** Heatmap summarizing the copy number abnormalities of unique Vk*MYC MM, included in the GISTIC analysis. **C** Example molecular time analysis for the indicated clonal chromosomal gain in one representative Vk*MYC mouse. The confidence of interval were generated using bootstrap function in the mol_time package (https://github.com/UM-Myeloma-Genomics/mol_time). **D** Molecular time comparison between Vk*MYC mouse (n = 19) and human MM (n = 21). P values were estimated using two-side Wilcoxon text. Boxplots display the median, quartiles, and variability of the data. **E** Percentage of male and female Vk*MYC MM with an M-spike greater than 7 g/L, measured every 10 weeks by serum protein electrophoresis. The number of mice analyzed is indicated, as well as the log-rank P values calculated by the log-rank (Mantel-Cox) test.

duplicated segment across 19 cases (i.e., the corrected ratio between duplicated and non-duplicated mutations)[12,23]. Similarly to what observed in MM, in particular in hyperdiploid cases, large trisomies in the Vk*MYC myeloma tended to be acquired within the same time window in 8/10 (80%) cases where this analysis was possible (Fig. 2C; Supplementary Fig. 7, Supplementary Data 12, 13). However, the Vk*MYC MM molecular time estimates were significantly higher compared to that observed in human MM, suggesting a later acquisition in disease pathogenesis (p < 0.00001 using Wilcoxon-test; Fig. 2D; Supplementary Data 13). Notably, in 4 out of 10 cases with multiple gains, we detected indications of at least two-time windows among clonal gains, underscoring clear evidence of independent and spontaneous acquisition of genomic events over time (Vk12598, Vk31159 and

Vk38525; Supplementary Fig. 7). Overall, our unsupervised CNV analysis identified several similarities between Vk*MYC and human MM, including recurrent events on distinct driver genes and the simultaneous acquisition of large chromosomal duplications and deletions.

## Structural variants and complex events
Using WGS (n = 41) and mate-pair WGS (n = 11) we perform a comprehensive characterization of SV in Vk*MYC MM (Supplementary Data 1, 2, 14). After removing the Vk*MYC transgene insertion breakpoints and the deletions associated with VDJ and class switch recombination within immunoglobulin genes (Ig), we observed a median of 8.5 (range 0–351) SVs per tumor, significantly lower compared to that observed in human MM (16, range 0–351; p = 0.00017 using Wilcoxon test).

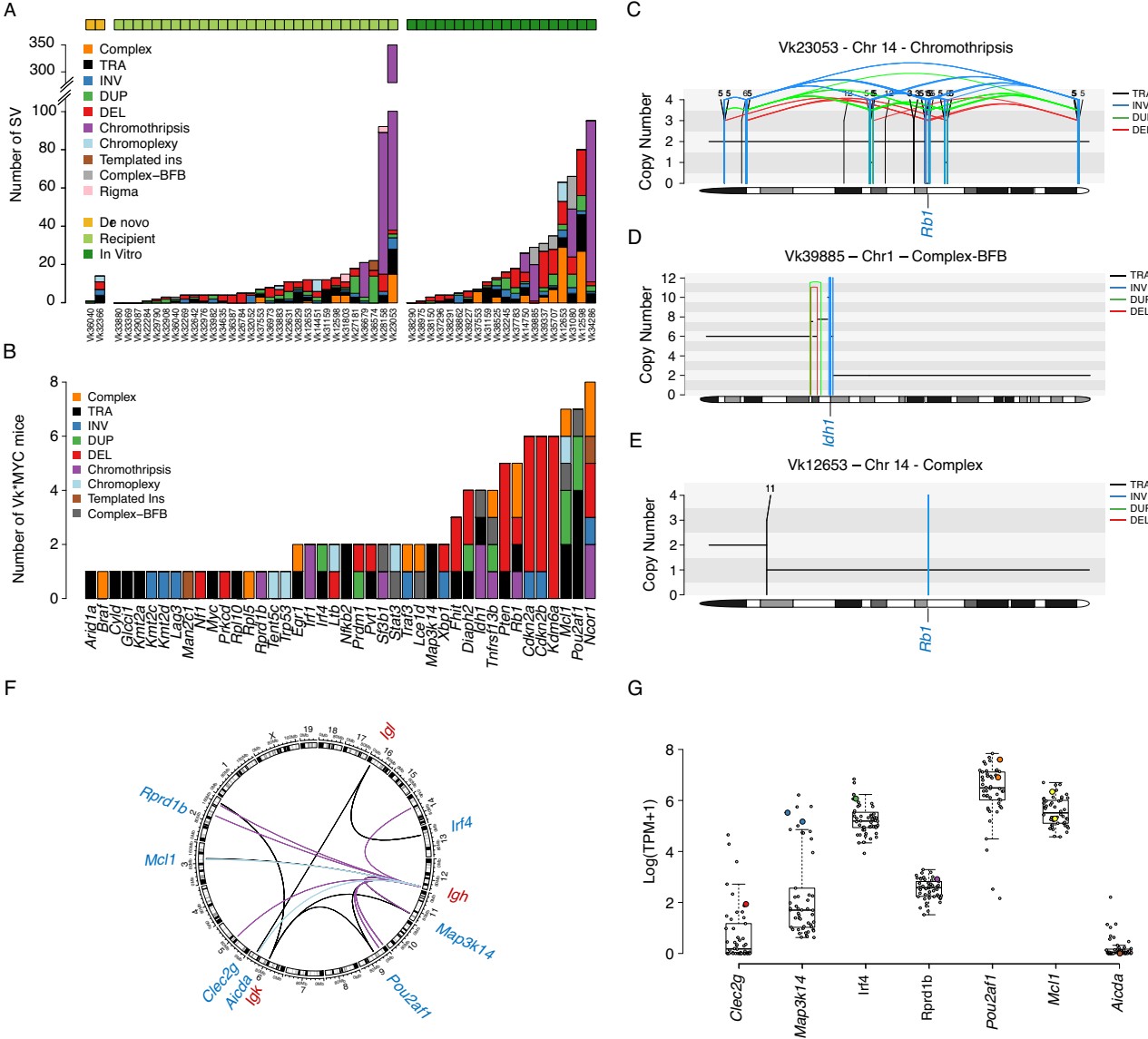

**Fig. 3 | Vk*MYC structural variants (SV) landscape. A** Barplot summarizing the prevalence and distribution of SV and complex events across 52 Vk*MYC MM. **B** Number of Vk*MYC MM mice with SV involving key oncodrivers. **C–E** Representative examples of SV events involving oncodrivers. BFB: breakage-fusion-bridge. The horizontal black line indicates the total copy number; the dashed orange line indicates the minor copy number. The vertical lines represent SV breakpoints: black: translocations; red: deletions, green: tandem-duplications; blue: inversion. **F** Circus plot showing all the immunoglobulin translocations detected. **G** Impact of immunoglobulin translocations on the partners' gene expression (colored dots). Boxplots display the median, quartiles, and variability of the data. A total of 50 Vk*MYC WGS with paired RNAseq was included in this analysis.

Interestingly, evidence of spontaneously acquired chromothripsis, chromoplexy, templated insertion, and complex not otherwise specified SV were observed in 13%, 7.7%, 2%, and 23% of mice, respectively (Fig. 3A–D)[18]. No differences were observed in SV burden and prevalence of different SV types and complex events among the different stages (Supplementary Data 15).

Complex and single SV events involved several oncodrivers including *Cdkn2a/Cdkn2b, Kdm6a, Tnfrsf13b, Rb1, Ncor1* and *Mcl1* (Fig. 3B–E). Interestingly, we observed complex events with evidence of breakage-fusion-bridge inducing multiple focal gains of different oncogenes such as *Mcl1, Stat3, Pou2af1* and *Idh1* (example shown in Fig. 3D). As additional similarity between human and Vk*MYC MM, translocations involving the immunoglobulin heavy (seven tumors) and light chain (five tumors) loci were detected in nine individual Vk*MYC MM (15.4%) (Fig. 3F). Of the translocations into the heavy chain locus, the ones affecting *Map3k14, Aicda* and *Traf3* are located

within the switch region, while the ones affecting *Pou2af1* and *Mcl1* are not (Supplementary Data 16). Some of these events partnered with known oncogenes promoting their overexpression (e.g., *Map3k14, Irf4, Pou2af1,* and *Mcl1*; Fig. 3G). Two translocations are within the introns of partner genes *Aicda* and *Traf3*, both with very low or undetectable expression (Fig. 3F, G). Again, no differences in driver gene involvement by either SV or by immunoglobulin translocations were observed among the stages of MM progression (Supplementary Data 15).

Overall, while variations exist in the prevalence of specific complex events and the participation of distinct oncodrivers, the Vk*MYC SV landscape exhibits intriguing similarities with human MM. Notably, these shared features include occurrences of chromothripsis and translocations involving the immunoglobulin genes, underscoring the relevance of the Vk*MYC model as a valuable tool for studying MM pathogenesis. In contrast to human MM, Vk*MYC tumors lack IgH translocations to *Ccnd* or *Maf* gene families or *Nsd2/Fgfr3*.

## Retrotransposition of Intracisternal-A-Particle (IAP) LTR mediates gene dysregulation

We have previously reported that retrotransposition of an IAP LTR-mediated ectopic expression of *Map3k14* in Vk12653 MM[24]. Using RNAseq we subsequently detected additional Vk*MYC samples with outlier expression of *Map3k14*, but our pipeline identified a relevant SV involving IgK and IgH in only two of them (Fig. 3F, G). We therefore visually inspected the BAM files of Vk*MYC MM samples looking for evidence of IAP LTR insertion within the *Map3k14* gene. In 13 tumors, *Map3k14* transcription initiated from an ectopic IAP LTR upstream of exon 2 or 3. When available, WGS was able to exactly map the insertion site. (Fig. 4A, Supplementary Data 17). In all cases with both RNAseq

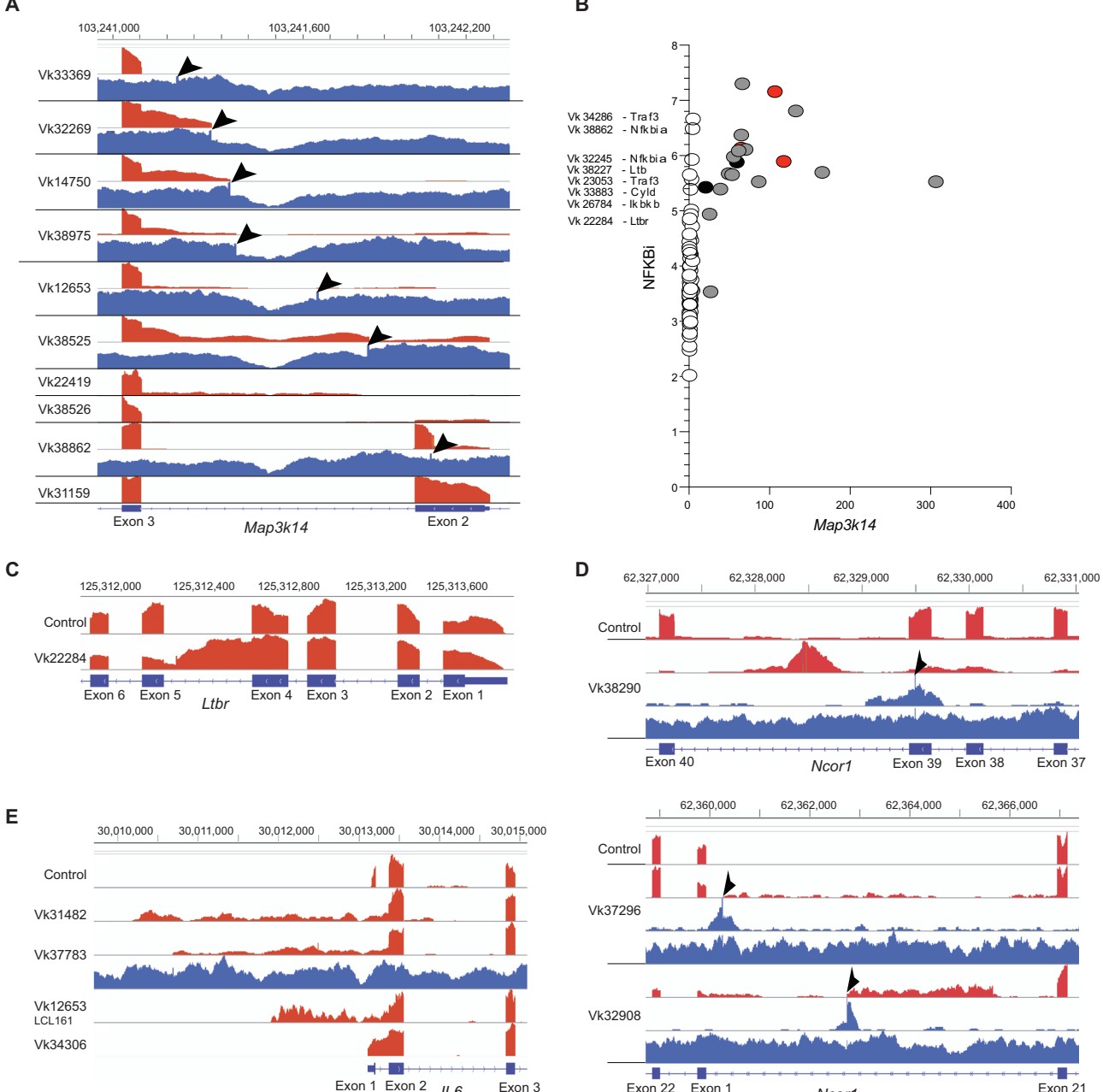

**Fig. 4 | LTR retrotransposition mediated gene dysregulation. A** Read depth for RNA (red) and whole genome (blue) sequencing of Vk*MYC tumor lines for Map3k14 is shown. By sequence analysis the 5'end of the RNA for these samples originates in an IAP LTR. The IAP insertion site in the DNA is marked by a 6–8 nucleotide duplication indicated by the arrowheads. The RNA for Vk31159 upstream of exon 2 originates from IAP LTR sequences, and WGS identifies an IAP insertion site at chr11:103253336 in intron 1 (not shown). **B** Gene expression NFkB index plotted versus Map3k14 expression (RPKM). Each dot represents an individual tumor; in grey are highlighted those with a mapped IAT insertion, in red those with an Ig translocation and in black those with unaccounted Map3k14 overexpression. The mutated gene in other samples with high NFkB index are listed. **C** Read depth for RNA (red) for *Ltbr* in Vk22284 with arrowhead highlighting the 6 nucleotide duplication at the IAP insertion site **D** Read depth for RNA (red), discordant reads from whole genome (upper blue), and all reads from whole genome (lower blue) sequencing of Vk*MYC tumor lines is shown. The IAP insertion site in the DNA is marked by a 6–8 nucleotide amplification indicated by the arrowheads, and is most easily seen visualizing the discordant reads only. Exon numbering for Ncor1 is from reference transcript NM_011038. The exon labelled 1 is an alternatively spliced exon of Ncor1 not included in any reference transcripts. It has also been identified as the first exon of Rip13a/Ncor1, an Ncor1 isoform lacking repressor domains. **E** Read depth for RNA (red) and whole genome (blue) sequencing of Vk*MYC tumor lines for Il6 is shown. By sequence analysis the 5'end of the RNA for these samples originates in an IAP LTR. The IAP insertion site in the DNA is marked by a 6-8 nucleotide amplification indicated by the arrowheads.

and WGS, outlier expression of Map3k14 could be accounted for by either Ig translocation or IAP insertion. In two cases, lacking WGS, the overexpression of *Map3k14* remained unaccounted. Altogether 17/88 (20%) of the MM tumors analyzed had outlier expression of *Map3k14*. The majority of the LTR insertions result in the elimination of exon 2 encoding the TRAF3 interacting domain known to be important for Map3k14 protein stability, as we and others previously noted in human MM, and cause constitutive NFkB activation, reflected by a high NFkB index by gene expression (Fig. 4B)[25,26]. Similarly, high NFkB index was observed in other Vk*MYC MM cases carrying mutations in NFkB regulatory genes described above (i.e., *Nfkbia*, *Ltb*, *Traf3*, *Ikbkb*, *Cyld*, Fig. 4B). We also identified an IAP insertion in *Ltbr* in Vk22284 resulting in high-level *Ltbr* expression, and an elevated NFkB index (Fig. 4B, C).

We performed an unbiased screen of our WGS using RetroSeq to identify other genes with recurrent insertions of LTR[27]. We identified 38 genes with more than two intragenic LTR insertions in the samples analyzed. Of the genes we have identified as recurrently mutated in MM, in addition to *Map3k14*, we found *Ncor1* to have IAP LTR insertions disrupting its expression (Fig. 4D, Supplementary Data 17).

Finally, the Balb/c plasmacytoma line MPC11 has been reported to have an IAP LTR insertion upstream of *Il6* resulting in its upregulated expression[28]. Expression of *Il6* was detectable by RNAseq in only four Vk*MYC MM, which, in all cases, we determined originated from an IAP LTR (Fig. 4E, Supplementary Data 17).

## Shared pathways of tumor progression between murine and human MM

Combining the results from the genomic and transcriptomic analyses across all different somatic events spontaneously acquired during Vk*MYC myeloma pathogenesis, we noted the convergence of acquired mutations on pathways activating NFkB (38%), RAS/mTORC1 (27%), cell cycle (48%) and chromatin modifiers (67%) as observed in human MM (Fig. 5A)[13]. We identified classic RAS activating mutations in only two samples (*Nras* Gln61Lys in Vk29790 and *Kras* Gln61Arg in Vk32829).

As tumors progress from de novo, to transplant to in vitro stage, we observed an increased genomic complexity associated with a higher gene expression proliferation index (GPI), a known adverse event in human MM[29]. Remarkably, in both human and murine PCs, GPI inversely correlated with immunoglobulin expression: Ig transcription progressively declined from normal PCs, to MGUS – or monoclonal gammopathy described in two recent mouse models based on *Ikk2ca* activation in germinal center B cells -[30,31], to newly diagnosed MM – or de novo Vk*MYC or the *Ikk2ca* crosses -, to relapse/refractory MM – or transplant Vk*MYC or 5TGM1 MM - to human or murine PC lines that grow in vitro, which generally lack Ig expression (Fig. 5B, C). We also observed a similar increase in NFkB index associated with the progression of both human and mouse MM (Supplementary Fig. 8). A tabulation of all the main molecular and behavioral features of each of the 58 established and characterized Vk*MYC transplantable and 25 *vitro* lines is provided (Supplementary Data 18, 19)

## Vk*MYC MM tumors remain dependent on MYC expression

Considering the progressive accumulation of genomic abnormalities associated with tumor progression, we wondered if Vk*MYC MM remained dependent on MYC dysregulated expression, which in this model is the driver of progression from monoclonal gammopathy. With this question in mind, we designed the original Vk*MYC transgenic construct with two LoxP site flanking the transgenic 3′ kappa enhancer, enabling its floxing upon CRE-mediated recombination, with consequent loss of MYC expression (Fig. 5D). We subsequently generated a derivative Vk*MYCΔloxP strain, in which the two LoxP sites have been removed, to retain MYC expression in the presence of CRE recombinase. We crossed both Vk*MYC and Vk*MYCΔLoxP mice with a strain carrying a tamoxifen (TAM) inducible CRE allele in the ROSA26 locus (Supplementary Data 3). We aged double transgenic mice until

they developed MM, and from them established MYC⁺CreERT2⁺ transplantable cell lines: Vk22284 Vk*MYC⁺CreERT2⁺ and Vk21153 Vk*MYCΔLoxP⁺CreERT2⁺ (Supplementary Fig. 9A–C). Finally, we treated MYC⁺CreERT2⁺ Vk22284 and Vk21153 tumor-bearing mice with tamoxifen for five consecutive days and monitored M-spike levels. While CRE induction had no consequences on Vk21153 MM growth, we observed a rapid reduction of M-spikes in Vk22284 tumor-bearing mice, indicating continuous dependency on MYC expression for MM survival (Fig. 5E). Tumors eventually relapsed despite retreatment with TAM; their molecular analysis showed that all contained an unfloxed VkMYC allele, highlighting the selection of MYC expression for tumor growth (Supplementary Fig. 9D, E).

## VK*MYC MM SBS signatures landscape

The genome-wide mutational landscape of newly diagnosed MM in humans is shaped by seven different SBS signatures: SBS1 and SBS5 (aging), SBS2 and SBS13 (APOBEC), SBS9 (poly-eta in the germinal center), SBS18 (radical oxygen stress damage), and SBS8. Performing mutational signatures analysis on all VK*MYC MM with available WGS, we extracted the same seven SBS signatures detected in human MM (Methods; Fig. 6A; Supplementary Fig. 10A, B; Supplementary Data 20). Additionally, SBS17, of unknown etiology, was detected in all but four Vk*MYC MM ($n = 90\%$). The presence of SBS9 was also observed in all cases, in line with the post-GC origin of the Vk*MYC PC. Looking at the Ig loci and AID-off target genes, we observed clear evidence of somatic hypermutation and SBS84 (AID, a mutational process usually undetectable in mutational signature genome wide analysis) and SBS9 involvement in all but five MM (Fig. 6B), confirmed by a comparative and focused analysis restricted to the clonal Ig genes in murine and human PCs (Fig. 6C). SBS84 localized mutational activity and AID involvement was also observed across known AID off-target genes, similar to what has been reported in human MM, and implicated in the mutations affecting the driver genes: *Dusp2, Pim1, H1f4/Hist1h1e, and H1f2/Hist1h1c* (Supplementary Fig. 10). Consistently, we confirmed targeting of the somatic mutation process to the Vk*MYC transgene causing reversion of the stop codon that controls MYC translation in 71 out of 84 unique MM analyzed by mRNA, associated with an average of 4.5% somatic hypermutation at the immunoglobulin loci[7], while the remaining cases without reversion of the stop codon have less than 2% (Supplementary Data 18, 19). We are actively investigating the mechanism of transformation in the cases without reversion and will report the results separately. Moreover, the majority (93.3%) of Vk*MYC MM tumors, like human MM, have undergone class switch recombination to express IgG or IgA; and out of the five cases expressing IgM, four have less than 2% somatic hypermutation at the immunoglobulin loci and lack reversion of the transgenic stop codon, suggesting a germinal center independent tumor origin, or an early germinal center exit, as recently reported in MM that develops in various different crosses with Ikk2ca activated by IgG1-CRE[30,31]. In line with this, the proportion of SBS9 was significantly lower in IgM Vk*MYC mice compared to the others ($p = 0.002$ using Wilcoxon test). Interestingly, we noted a shift in the Vk*MYC predominant isotype over time from IgG[7] to IgA, that we suspect is related to a change in the microbiome occurring in our colony (Fig. 6D)[32].

Most interestingly, APOBEC mutational activity was detected in 44% of Vk*MYC MM by WGS (Fig. 6A, E) and, when present, its relative contribution was similar to that of human MM (Fig. 6F). Although this proportion is lower in comparison to the 80% we have previously reported for human MM and its progressive precursor conditions[6], the significant discovery that Vk*MYC MM spontaneously develops APOBEC mutagenesis is a critical finding, considering that no other mouse model of MM has been reported to spontaneously acquire an APOBEC mutational signature (see Methods)[30,31]. Importantly, no correlation was noted between the presence of APOBEC and AID signatures, in line with the distinct nature and role of these two deaminases

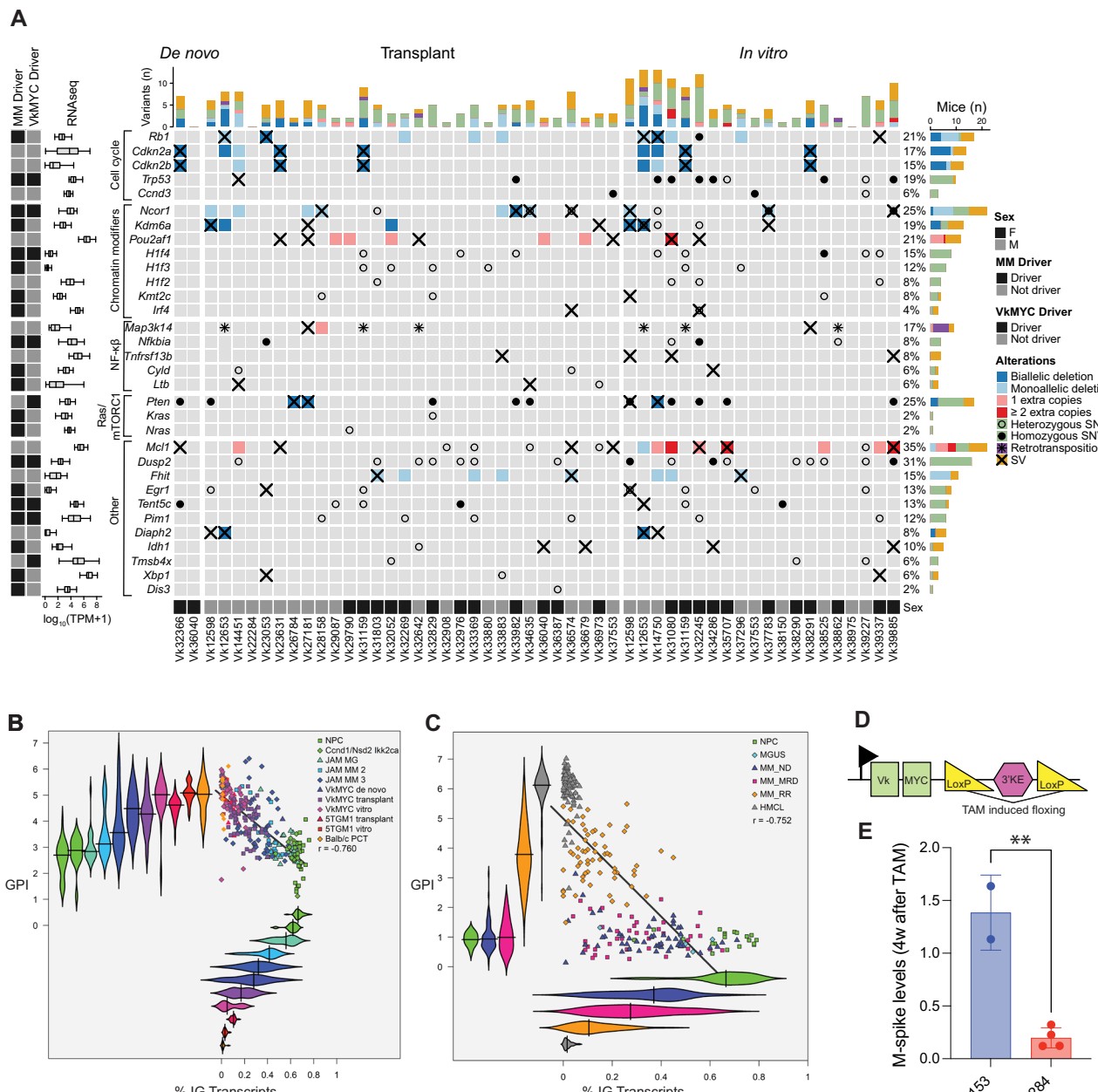

**Fig. 5 | The genomic landscape of the Vk*MYC mouse model of MM. A** Heatmap summarizing all the key oncodrivers involved by somatic events in the Vk*MYC MM. Only CNV involving MM oncodrivers and with a length smaller than 3.5 mb are reported. **B–C** XY scatterplot of percent of immunoglobulin transcription versus gene expression proliferation in murine, **B**, and human, **C**, plasma cell tumors. The % of IG transcription is derived from RNASeq and the GPI score is the mean of the log2 transformed TPMs of the 50-genes comprising GPI29. The Pearson correlation score is indicated. **D** Graphic representation of the Vk*MYC construct (not to scale).

Green squares represent the kappa variable region and human MYC exons. The position of the two LoxP sites flanking the 3′ Kappa enhancer is shown, as well as their recombination following tamoxifen-induced CRE expression. The horizontal arrow indicates the transcription start point. **E** M-spike levels measured four weeks after tamoxifen treatment and normalized to day 0 levels in mice bearing Vk21153 Vk*MYCDLox/CreERT2 and Vk22284 Vk*MYC/CreERT2 MM tumors. Boxplot is presented as mean values +/− standard deviation. ** Indicate the two-tailed unpaired t test *P* value = 0.0022.

(Supplementary Data 20). To validate the presence of APOBEC mutational signature (SBS2) in Vk*MYC MM, we analyzed the SBS signatures on the WES cohort, identifying APOBEC mutational activity in 52% of cases (Supplementary Fig. 11A). Similar to human MM, APOBEC contribution tends to increase from clonal to subclonal variants, suggesting a late role in Vk*MYC MM development (Fig. 6G). Investigating the five Vk*MYC MM collected at different stages with available either WGS or WES, we observed APOBEC mutational activity in both samples in 4/5 cases. In Vk36040, the transplantable WGS showed SBS13, while the de novo sample did not show any evidence. Based on the

concordance of all the other paired samples, APOBEC mutagenesis was likely already active but undetectable at the de novo stage, however this cannot be established with confidence with current bulk sequencing data. Additionally, *Apobec3* and *Apobec1* expression were confirmed using RNAseq data from 89 cases (Supplementary Fig. 11B). No difference in APOBEC mutational contribution was observed between different states, low and high *Apobec3* expression, and *Apobec3* mutated cases. To further investigate the APOBEC role and activity in VK*MYC MM, we interrogated whole BM single-cell RNAseq data generated from 15 previously published Vk*MYC mice (114,363 total

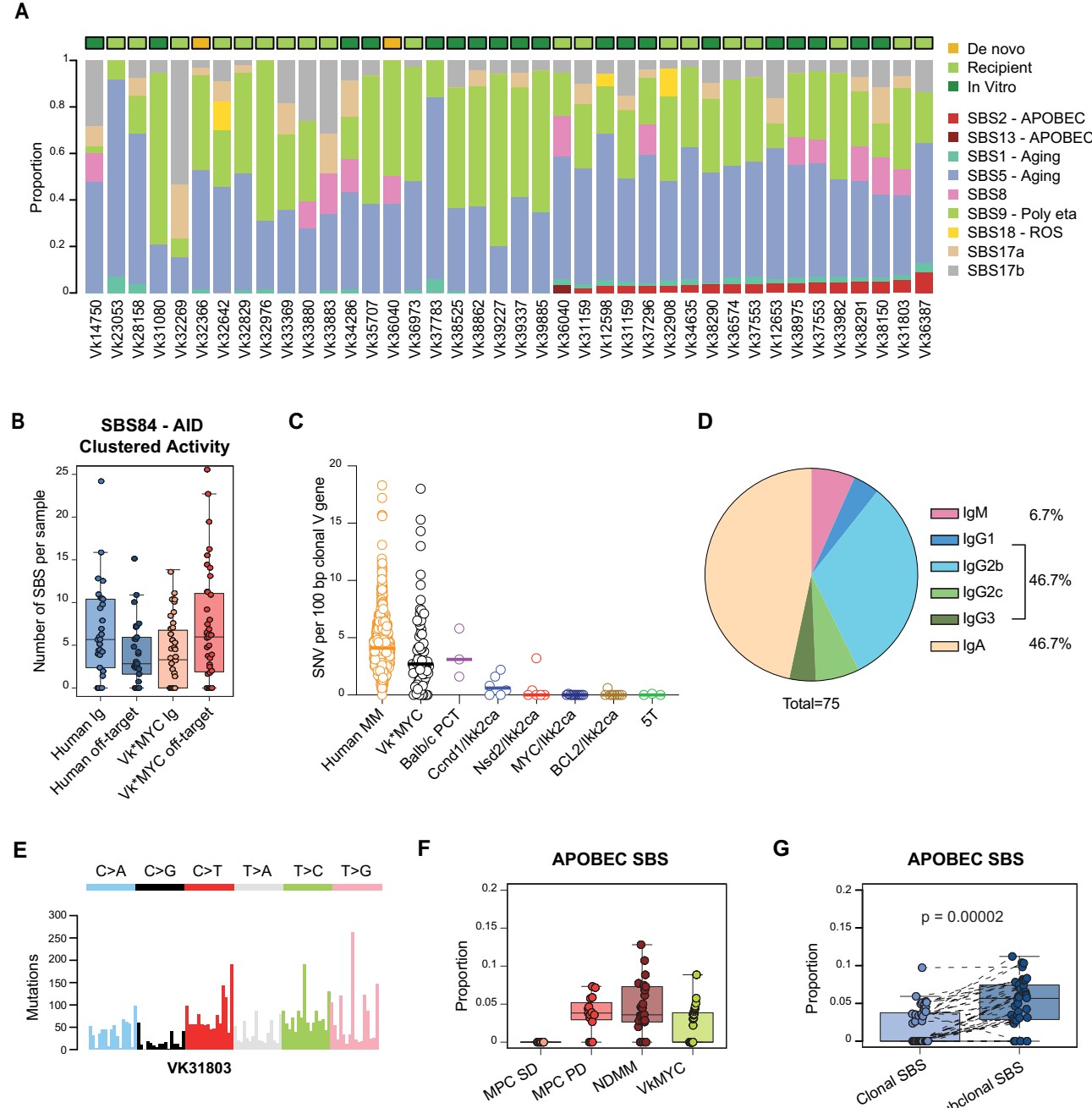

**Fig. 6 | Vk*MYC mouse MM Single base substitutions (SBS) mutational signatures landscape. A** Barplot showing the contribution of each mutational signatures for each WGS. ROS: radical oxygen stress. **B** Number of SBS84 (AID) clustered mutations in human (*n* = 30) and mice (*n* = 41) involving either immunoglobulin (Ig) or off-target genes. **C** Percentage of somatic mutation (SNV per 100 base pair) at the productive immunoglobulin allele across tumor types. Each circle represents an individual tumor; in pink are highlighted IgM expressing Vk*MYC MM tumors. **D** Distribution of immunoglobulin isotypes across Vk*MYC MM tumors. **E** Example of a 96-classes profile from a Vk*MYC mouse MM with clear APOBEC mutational activity. **F** Proportion of mutational signature due to APOBEC across Vk*MYC mouse (*n* = 41) and human MM (*n* = 30) and precursor conditions. MPC SD: stable myeloma precursor conditions (*n* = 13); MPC PD (*n* = 17): progressive precursor conditions. For human we included previously published data by Oben et al. Nat Comm 2021. Boxplot is presented as mean values +/− standard deviation. **G** Proportion of mutational signature due to APOBEC across clonal and subclonal SBS in Vk*MYC mouse MM. Two-side *p*-values were estimated using paired Wilcoxon text. Boxplot is presented as mean values +/− standard deviation.

cell count, Supplementary Fig. 12 and Fig. 7)[10]. *Apobec3* was expressed at single cell level by both, the tumor cells and by normal B-cell and normal plasma cells (Fig. 7E, F). In contrast, *Apobec1* expression was mostly restricted to the tumor plasma cells (Fig. 7C, D). No significant difference in the mutational contribution of neither APOBEC nor other mutational signatures were observed across different stages (Supplementary Fig. 13). The fact that Vk*MYC mice develop plasma cell

tumors with spontaneous aberrant activation of APOBEC further support the similarities in the molecular pathogenesis of disease progression between VK*MYC and human MM.

## Discussion

In this study, we performed a comprehensive characterization of the genomic and transcriptomic landscape of Vk*MYC MM, which includes

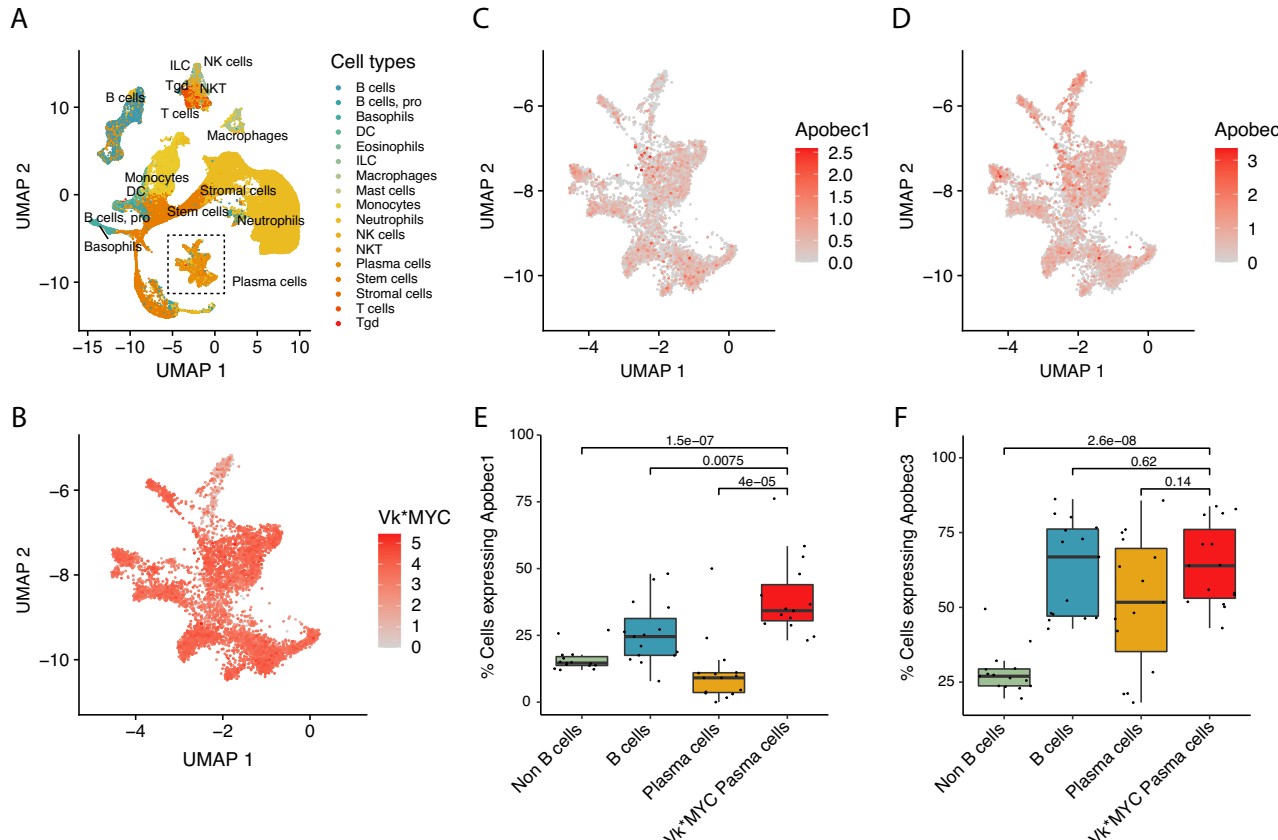

**Fig. 7 | Single cell RNA expression of Apobec1 and Apobec3 across different immune populations in the Vk*MYC mouse. A** UMAP analysis of all cells analyzed color coded by cell type. **B** Expression of Vk*MYC transcript in plasma cells. **C–D** Expression of Apobec1 (**C**) and Apobec3 (**D**) in plasma cells plasma cells.

**E–F** Difference in Apobec1 (**E**) and Apobec3 (**F**) expression a cross different cell populations. Two-sided *p* values were estimated using the Wilcoxon test. Boxplots are presented as mean values +/− standard deviation.

39 original de novo tumors from aged transgenic mice, 63 transplantable tumors lines that can be passaged in vivo into non-irradiated, syngeneic C57Bl/6 immunocompetent recipients, and 25 tumors capable of growing in vitro. The integration of WGS, WES, bulk and single-cell RNA data, revealed genomic similarities between Vk*MYC and human MM not just based on individual driver genes or dysregulated pathways but also on broader genomic features, such as presence of chromothripsis, large simultaneously acquired trisomies, and APOBEC mutational activity. Supplementary Data 21 summarizes similarities and differences between Vk*MYC and different subtypes of human MM.

Overall, Vk*MYC MM shares some genetic features with the 10% of human MM having MYC translocations and CCND2 expression but lacking primary recurrent Ig translocations. This group is primarily comprised of a subset of HRD lacking trisomy 11 (HRD11-) MM, but also contains a fraction of patients that lacks hyperdiploidy and has high frequency of NFkB activating mutations, which has been classified as nHRD2[14,25,33–36].

Unique to Vk*MYC MM is the hijacking of a murine-specific mutational process that utilizes the IAP transposable element to dysregulate *Map3k14*, *Ltbr*, *Il6* and *Ncor1*, highlighting the central role of the NFkB, STAT3 and chromatin modifier pathways in MM pathogenesis. IAP expression has been reported to be characteristically high in BALB/c plasmacytomas, with sporadic oncogene activation by retrotransposition[37]. Here, we report three recurrent sites for spontaneous oncogenic insertional mutagenesis by an endogenous retroelement. Retrotransposition is much more common in mouse than human likely because mice have a larger group of active endogenous retroviral elements, including IAP. In contrast, humans have only a few

active transposable elements (TE), mainly LINEs and SINEs. Genome-wide analysis of human TE has been conducted in multiple cancer, but in contrast with solid tumors, hematological malignancies rarely present these genomic events[38]. Another striking difference between Vk*MYC and human MM is the higher prevalence of mutations of *Dusp2* (a negative regulator of *Stat3* pathway[39]), *Pim1*, *Kdm6a*, *Ncor1* and *Pten* (a negative regulator of mTORC pathway) and low incidence of mutations of *Ras* (an activator of MAPK and mTORC1[40] pathways) in the mouse.

APOBEC mutational signature is a key genomic feature identified in MM. APOBEC deaminases are known to introduce hundreds of SBS in different cancers, but its aberrant activity is virtually undetectable in normal mouse and human cell genomes[41]. APOBEC mutational activity has emerged as the most sensitive and prevalent MM-defining genomic event associated with MM precursor condition progression[6,13,42]. While *MAF/MAFB* translocated MM patients are characterized by a high APOBEC mutational contribution (i.e., hyper-APOBEC), mostly driven by APOBEC3A isoform, 80% of MM and progressive MM precursor conditions have a lower APOBEC mutational activity where the isoforms 3A and 3B play an equal role (i.e., canonical APOBEC)[13,43,44]. Despite its high prevalence in different cancer types[43], and recent data suggesting a link with cancer cell immune-escape, genomic instability, and systemic seeding[45,46], the oncogenic role of canonical APOBEC is largely unknown. Moreover, most of the investigations used models in which APOBEC is already constitutively active and mostly driven by the APOBEC3A (hyper-APOBEC)[47,48], therefore failing to capture the spontaneous canonical APOBEC activation is seen in MM and lymphomas[49]. In this context, the ability of the immunocompetent Vk*MYC mouse model to develop a plasma cell tumor with canonical

and spontaneous APOBEC mutational activity both represents a confirmation of the genomic similarities between human and Vk*MYC mice MM, as well as a unique model system to explore how and why APOBEC activate during cancer development. Interestingly, leveraging scRNAseq of the entire Vk*MYC plasma cells and surrounding microenvironment, we observed a significant expression of *Apobec3* and a clear enrichment of *Apobec1* compared to other normal population. While *Apobec3* mutational activity has been demonstrated in mice, *Apobec1* emerged as tumor plasma cell-specific potential promoter of SBS2 and SBS13.

We believe the biological fidelity of the Vk*MYC model to human MM is attributable to its unique design. It is based on the introduction of a *MYC* transgene in the C57BL/6 strain that spontaneously develop monoclonal gammopathy[50]. Remarkably, when backcrossed to a Balb/c strain that does not spontaneously develop gammopathy, Vk*MYC mice fail to develop MM, suggesting that *MYC* activation in a permissive genetic background provides the conditions for a transition from a stable to a progressing monoclonal gammopathy[35]. Despite *MYC* activation, Vk*MYC mice remain asymptomatic for at least a year before acquiring MM clinical features, and full MM progression in Vk*MYC mice requires a long latency (70 weeks on average), allowing for the spontaneous acquisition and selection of additional mutations. As a result, each individual Vk*MYC MM displays a unique genomic profile. The heterogeneity and genomic similarity with human MM create an opportunity for future investigations aimed to longitudinally track the genomic and tumor microenvironment evolution of Vk*MYC MM over time that could inform on key drivers of the progression from SMM to MM human. These types of investigations would not be feasible in humans due to the long time frame and ethical considerations. In is important also to highlight that the Vk*MYC spontaneous genomic evolution and heterogenicity are in contrast to other genetically modified mouse models of cancer, where engineered expression of multiple oncogenes limits the selection for genomic diversity[30,31]. Prior studies identified continued MYC-dependence using mouse models of lymphoma and leukemia initiated by MYC[51]. Here, we show the same holds true in a mouse model of MM, which may in part explain the marked therapeutic efficacy of drugs targeting MYC in the treatment of MM, such as IMiDs.

Interestingly, the genomic life-history of the Vk*MYC MM pathogenesis shared similar patterns with what is observed in human MM. Specifically, MM in both species is characterized by progressively increased proliferation and decrease in immunoglobulin expression, APOBEC mutational activity, mutations in distinct driver genes (e.g. *TP53*), and by the intermediate acquisition of chromothripsis, structural variants, and focal copy numbers[12,18,52]. Another shared temporal similarity is that both human and Vk*MYC MM exhibit a predominant influence of CNV and SV in their initial stages, followed by the acquisition and subsequent selection of SNV/indels in driver genes during later phases. In contrast, certain events co-occurred in both species, but with an opposite timing. In human MM the simultaneous acquisition of large chromosomal gains is usually one of the earliest events, and *MYC* translocation occurs as one of the latest events driving the final progression into symptomatic MM. In contrast to this timeline, *MYC* activation is the earliest events in Vk*MYC MM, and the simultaneous acquisition of large chromosomal gains is usually acquired later in time[12,13,36,53,54]. Despite this opposite timeline, our data suggest that the overexpression of *MYC*, alteration of the NFkB pathway and large chromosomal gains are essential for the development of a large fraction of PC tumors in both the Vk*MYC mouse and in humans, as also recently confirmed in another GEMM of MM[30].

Vk*MYC MM cell lines Vk12598 and Vk12653 have been extensively used for myeloma research. Some of the advantages they offer are a high level of immunoglobulin transcription, dependence on the in vivo microenvironment, rapid engraftment in immunocompetent C57Bl6 mice and fidelity to human MM. The additional cell lines we are reporting here, with their extensive genomic characterization, greatly expands the repertoire of unique MM tumor models that may be used to reflect different aspects of MM biology and we hope will be a rich resource for investigators in the field for years to come.

In summary, through comprehensively genomic profiling of plasma cell DNA and bulk and single-cell RNA, the Vk*MYC mouse emerges as a model that recapitulates several key genomic features observed in human MM (Supplementary Data 21). As a result, the Vk*MYC model provides a unique opportunity in future research to uncover the step-wise genomic events within plasma cells not otherwise feasible in human studies to be able define the genomic underpinnings of MM disease progression. Finally, this study highlights the importance and advantage of applying comprehensive genomic investigations in mouse models in which genomic events are spontaneously acquired over time. Detailed knowledge of the spontaneous cancer evolution in such models has major potential in deciphering the early phase of cancer initiations, the selection/acquisition of distinct genomic events in the context of an immunocompetent environment, creating robust biological rational for effective pre-clinical interventions.

## Methods

### Vk*MYC mouse model and cell lines

The animal care and use program and facilities at Mayo Clinic Arizona meet all federal regulations and guidelines. Mayo Foundation maintains a NIH animal welfare assurance statement (A3291-01) with the Office of Laboratory Animal Welfare (OLAW) and is registered with the United States Department of Agriculture (USDA) (41-R-0006) as an animal research facility. Mayo Clinic Arizona's animal care and use programs have been reviewed and fully accredited by the Association for Assessment and Accreditation of Laboratory Animal Care International (AAALAC) (#000880). All experiments were performed under the approval of the Mayo Foundation Institutional Animal Care and Use Committee (protocol A00001948-16-R22) and conformed to all the regulatory environmental safety standards. Mice were housed – 5 mice per cage - in a conventional facility in Allentown NexGen ventilated mouse racks with NuAire change stations in each room and fed either a Purina 5053 standard rodent chow or Purina 5058 breeder rodent chow. Reverse osmosis purified municipal chlorinated and fluoridated water was provided. Light cycle was 10–14 10 h of darkness (off at 7:00 pm) and 14 h of daylight (on at 5:00am) and temperature range was 68–79 °F, humidity range 30–70%. The generation and initial characterization of the Vk*MYC (RRID:MMRC_68098_MU), their derivative lacking LoxP sites, Vk*MYCDLox (RRID: MMRRC_068099-MU) and Vk*MYC^hCRBN mice and derived transplantable lines have been reported previously[7,55]. De novo mice of both sexes were aged and monitored for tumor burden by serum protein electrophoresis as previously described[9]. Briefly, M-spikes were quantified by calculating the ratio of densitometric values of the M-spike and albumin bands using the albumin reference value of 27 g/L[56]. Transplantable Vk*MYC lines mice were generated by serial transplantation (at least three times) of approximately 1 million unsorted tumor cells harvested from Vk*MYC de novo mice into 4–12 week old C57BL/6 recipient mice of both sexes[55]. Humane endpoints were based on IACUC established criteria: weight loss equal or greater than 20% body weight, inability to reach food or water, body condition score of 1. Euthanasia was performed by carbon dioxide inhalation. Vk*MYC *vitro* cell lines were generated by culturing permissive Vk*MYC cells harvested at the time on necropsy in RPMI-1640 + 10% fetal bovine serum supplemented with glutamine, penicillin, streptomycin and, when required, IL6. All cell lines were tested for mycoplasma contamination biannually using the MycoAlert kit (Promega) and were periodically validated by copy-number polymorphism by PCR. Floxing of the transgenic 3' kappa enhancer was achieved by treating tumor bearing mice with 1 mg tamoxifen (Sigma-Aldrich) in corn oil given by daily i.p. injection for five consecutive days. No statistical method was used to predetermine sample size.

## Whole genome, whole exome and mate-pair sequencing of VK*MYC tumor cells

Myeloma cells were harvested and purified by magnetic selection as previously described[55]. DNA was extracted from isolated plasma cells using Puregene (Qiagen). DNA similarly extracted from three Vk*MYC tails was used as normal match. For WES, paired-end libraries were prepared following the manufacturer's protocol (Agilent) using the Bravo liquid handler from Agilent. Briefly, 1 ug of genomic DNA was fragmented to 150–200 bp using the Covaris E210 sonicator. The ends were repaired, and an "A" base was added to the 3′ ends. Paired-end Index DNA adaptors (Agilent) with a single "T" base overhang at the 3′ end was ligated, and the resulting constructs were purified using AMPure SPRI beads (Agencourt). The adapter-modified DNA fragments were enriched by 4 cycles of PCR using SureSelect forward and SureSelect ILM Pre-Capture Indexing reverse (Agilent) primers. The concentration and size distribution of the libraries is determined on an Agilent Bioanalyzer DNA 1000 chip.

Whole exome capture was carried out using the Agilent SureSelect Mouse all exon kit. 750 ng of the prepped library was incubated with whole exon biotinylated RNA capture baits supplied in the kit for 24 h at 65 °C. The captured DNA:RNA hybrids were recovered using Dynabeads MyOne Streptavidin T1 (Dynal). The DNA was eluted from the beads and desalted using purified using Ampure XP beads (Agencourt). The purified capture products were then amplified using the SureSelect Post-Capture Indexing forward and Index PCR reverse primers (Agilent) for 12 cycles. The concentration and size distribution of the libraries was determined on Qubit (Invitrogen) and Agilent Bioanalyzer DNA 1000 chip. Exome libraries were loaded one sample per lane onto Illumina TruSeq v3 paired-end flow cells at concentrations of 9 pM to generate cluster densities of 600,000–800,000/mm² following Illumina's standard protocol using the Illumina cBot and TruSeq Rapid Paired-end cluster kit version 3. Some whole-exome libraries were sequenced on a NovoseqPE150 sequencer (Illumina) to generate 12 G of raw data per sample. Whole-genome libraries were prepared using the NEBNext® Ultra™ DNA Library Prep Kit (New England Biolabs) and sequenced on a NovoseqPE150 sequencer (Illumina) to generate 75 G of raw data per sample.

Nextera Mate Pair libraries were prepared following the manufacturer's protocol (Illumina). 1ug of genomic DNA in 76 ul EB buffer was simultaneously fragmented and tagged with a biotinylated mate pair junction adaptor. The resulting construct contained a short single stranded sequence gap, which was repaired enzymatically according to manufacturer's protocol (Illumina). The repaired DNA was purified, and smaller DNA fragments (<1500 bp) were removed using AMPure Beads. The size selected fragments were circularization by blunt end ligation for 16 hrs at 30 °C using circularization ligase (Illumina). Non-circularized fragments were eliminated by DNA exonuclease treatment. The remaining circularized DNA was again fragmented, this time using the Covaris E210, generating double-stranded DNA fragments with fragment sizes in the 200–2000 bp range. The biotinylated DNA fragments were purified using Dynalbeads M-280 streptavidin beads (Invitrogen) as outlined in the Illumina Mate-Pair protocol. Illumina-indexed adapters were added to the DNA on the M-280 beads using the TruSeq Library Sample Preparation kit (Illumina) as follows. The ends of the biotinylated fragments immobilized on the beads were repaired and phosphorylated using Klenow, T4 DNA polymerase, and T4 polynucleotide kinase; after which an "A" base was added to the 3′ ends of double-stranded DNA using Klenow exo- (3′ to 5′ exo minus). Paired-end DNA adaptors (Illumina) with a single "T" base overhang at the 3′ end were ligated and the immobilized adapter-modified DNA fragments are enriched by 10 cycles of PCR. The PCR supernatant was recovered from the beads using a magnetic rack. The PCR-enriched constructs were cleaned up with AMPure xp beads recovering DNA fragments of approximately 300–2000 bp. Concentration and size distribution of the libraries were determined on an Agilent Bioanalyzer

DNA 1000 chip and Qubit dsDNA assay (Invitrogen). Libraries were sequenced at 4 samples/lane to generate ~ 150 million reads/sample following Illumina's standard protocol using the Illumina cBot 3000/4000 PE Cluster Kit. The flow cells were sequenced as 150 ×2 paired-end reads on an Illumina HiSeq 4000 using TruSeq SBS sequencing kit version 3 and HCS v3.3.20 data collection software. Base-calling was performed using Illumina's RTA version 2.5.2.

Whole genomic and whole exome sequencing reads had adaptors trimmed using fastp v0.22 and were aligned the mouse reference genome (GRCm38/mm10) using BWA v0.7.17. Base quality score recalibration was performed using GATK v4.4 and duplicates were marked using MarkDuplicates. Somatic variants were called using Mutect2 and annotated using Annovar. Applying additional post-processing filters, only mutations with a ROQ score (i.e., phred-scaled qualities that alt allele are not due to read orientation artifact) >89, TLOD score (i.e., log 10 likelihood ratio score of variant existing versus not existing) >6, and DP (approximate read depth) score >5 were included in our analysis. Variants in known MM driver genes not passing these quality filters were manual inspected in IGV and were kept within the analysis if they did not appear to be an artifact. Additionally, we eliminated any WGS samples if there were less than 500 variants per genome or WES samples if there were less than 10 variants per exome. Driver gene discovery analysis was performed using *dndscv* R[11] package and the list inferred driver genes was not affected by the inclusion or exclusion of potential C57BL/6 wild-type short nucleotide polymorphisms (dnSNP) detected in single mouse.

CNV were called using the GATK4 Somatic CNV pipeline modified for mouse. GISTIC2.0 was used to estimate CNV enriched more than what it would be expect by chance. Focal CNV segments <10 kb with identical start or end positions in >3 mice were manually inspected and annotated as polymorphism artifacts and excluded from the analysis. All CNV calls are available in Supplementary Data 8.

SV were called using Lumpy[57] as implemented in Smoove release 19 (https://github.com/brentp/smoove) Complex SV were annotated by manual inspection following the previously published criteria[18]. SV gene involvement is defined by: the genes is within deletions and duplications <3 Mb, the genes is 100Kb from any SV breakpoint, the gene is up to 500Kb from a translocation or inversion. All SV calls are available in Supplementary Data 14.

Mutational signatures were estimated using first sigprofiler and *hdp* as de novo extraction, and then mmsig as fitting[58,59]. Molecular time analysis was run as previously described (https://github.com/UM-Myeloma-Genomics/mol_time)[13].

## aCGH of VK*MYC tumor cells

High-resolution aCGH was performed on Gentra-Puregene-cell-kit (Qiagen) purified DNA from 27 mice with the Sureprint G3 mouse CGH 244 K microarray kit (Agilent Technologies), as previously described[9]. The data have been submitted with GEO submission #GSE110954.

## Gene expression profiling

RNA from CD138-selected Vk*MYC plasma cells was extracted from TRIzol and further purified on Purelink Micro-Mini Columns (Invitrogen), with an On-Column DNAse Digestion Step. Gene expression profiling was performed on the Affymetrix mouse 430 2.0 array following the manufacturer's suggested protocol, as previously described (GSE111921)[9]. For RNAseq, mRNA libraries were prepared using NEBNext Ultra II RNA Library Prep Kit (New England Biolabs) with polyA enrichment and sequenced on a Novoseq S4 sequencer (Illumina) to generate 6 G of raw data per sample. RNA sequencing reads had adaptor sequences trimmed with fastp and were aligned to the mouse reference genome (CRCm38/mm10) using STAR v2.7.10b and transcript per million (TPM) gene expression values were obtained using Salmon v1.8.0. For comparisons between human and murine gene

expression, CoMMpass IA19 release was used—limited to bone marrow samples with available RNA-Seq data and either a 'baseline' or 'confirm progression' reason for visit. The unstranded, salmon TPM counts with immunoglobulin filtering were log2 scaled and used to derive the 50-gene published proliferation score (GPI) and 11-gene published NFkB Index (NFkBi)[26,29]. Details on immunoglobulin involvement were leveraged from RNA-Seq QC metrics. For mouse comparisons, immunoglobulin-filtered RNA-Seq TPM gene counts were similarly used. The GPI and NFkBi gene scores were computed from their respective mouse analogue genes. Clonal V(D)J recombinations were identified using MiXCR v3.0.3, and the percent identity of the dominant heavy and light chain variable genes to germline genes used to determine the degree of somatic mutation.

## Single-cell RNA sequencing of Vk*MYC bone marrow

scRNAseq data from a previously published study of unsorted Vk*MYC bone marrow mononuclear cells were downloaded from the Sequence Read Archive (SRP214856)[10]. BAM files were converted to fastq format and realigned to a custom version of the mm10 mouse reference genome (refdata-gex-mm10-2020-A) incorporating the Vk*MYC transgene using Cellranger v7.0 (10X Genomics)[10]. Gene count matrices and sample aggregation was performed using Cellranger. Cells were classified into cell type by SingleR v1.8.1 using the "immgen" reference dataset (Supplementary Fig. 12). Cell counts were normalized, variable features were found, and dimensionality reduction using PCA and UMAP were performed using Seurat v4.1.1. Plasma cells were identified by as those B cells classified by SingleR and having an average expression score >0 of selected plasma cell genes (*Sdc1, Tnfrsf17, Slamf7, Xbp1*) using the Seurat AddModuleScore function. Tumor plasma cells were identified as those cells classified as plasma cells and having a Vk*MYC transgene average expression score >0.

## Reporting summary

Further information on research design is available in the Nature Portfolio Reporting Summary linked to this article.

## Data availability

Raw WGS, Mate pair, WES and RNAseq datasets are available on NCBI Gene Expression Omnibus(RRID:SCR_005012) under accession number GSE255233 and on NCBI Sequence Read Archive (RRID:SCR_004891) under the NCBI BioProject accession number PRJNA938752. Publicly available data from previous mouse models were imported and reanalyzed following the pipeline described above: BioProjects PRJNA560057, PRJNA721176, PRJNA910238, PRJNA259862, PRJNA845532, PRJNA846769, PRJNA881497, PRJNA912227 and PRJNA759563[30,31]. A Source Data File is provided. Source data are provided with this paper.

## Code availability

No new code was generated in this study.

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

## Acknowledgements

This work was supported by the National Institute of Health (RO1CA234181, U54CA224018 and CA186781 MC, PLB, CKS, MES, MD, EWM), the Multiple Myeloma Research Foundation (MMRF), the Perelman Family Foundation, the Riney Family Multiple Myeloma Research Program Fund, and by a Sylvester Comprehensive Cancer Center NCI Core Grant (P30 CA240139). FM is supported by the American Society of Hematology, Leukemia and Lymphoma Foundation and International Myeloma Foundation. GJM received grant support through a Translational Research Program award from the Leukemia & Lymphoma Society (6020-20).

## Author contributions

Conceptualization: M.C., P.L.B., F.M. Data curation: F.M., M.C., P.L.B., D.C., E.B. Formal Analysis: F.M., M.C., P.L.B., D.C., E.B. Funding acquisition: F.M., M.C., P.L.B. Investigation: F.M., M.C., P.L.B., D.G.C., E.B., C.K.S., B.Z., M.E.S., M.T.D., E.W.M., Y.T.A., C.X.S., Y.X.Z., G.J.M., O.L. Writing – original draft: F.M., M.C., P.L.B., D.C. Writing – review & editing: all authors.

## Competing interests

MC and PLB receive royalties from Vk*MYC, Vk*MYC$^{hCRBN}$ mice and derivative transplantable lines. The remaining authors declare no competing interests.
