## [Peer Review File · Nature Communications]

The genomic landscape of Vk*MYC myeloma highlights shared pathways of transformation between mice and humansREVIEWER COMMENTS

Reviewer #1 (Remarks to the Author):

This manuscript by Maura et al. is a meaningful contribution and important resource for research in multiple myeloma (MM) and other plasma cell neoplasias. The Vk*MYC model and the different version of its tumor cells exhibit many features that faithfully recapitulate the pathophysiology of MM. Vk*MYC tumor cells (de novo, transplanted, or in vitro cultured) are extensively used to study MM biology of MM and response to treatments, so this detailed assessment of their genomic features is important. For this reason, it is especially important that the manuscript addresses more clearly some of its key questions, e.g. which stages in the "life history" of MM are recapitulated by the different versions of Vk*MYC tumor cells (de novo, transplanted, or in vitro cultured); whether the Vk*MYC model in general reflects many genomic subtypes of MM but primarily those cases which lack primary IgH translocations and if, by extension, the Vk*MYC model can be considered a model of "hyperdiploid MM". To their credit, the authors provide data for not only for those genomic features that are similar in Vk*MYC model and human MM, but also for some that appear distinct. Overall, this is an important resource for myeloma research and hopefully addressing the comments below will help enhance it.

Main comments:

1. It is important that the paper provides a clearer statement on which parts of the "life history of MM" are represented, based on the current genomic data, by de novo vs. transplanted vs. in vitro growing Vk*MYC cells.

Through the prior work of Drs. Chesi and Bergsagel on the Vk*MYC model, a widely held impression (perhaps incorrect?) in the MM field is that de novo Vk*MYC lesions simulate newly diagnosed MM; transplanted Vk*MYC lesions are reminiscent of relapsed and refractory MM; and Vk*MYC MM cells growing in vitro are the mouse equivalent of human MM lines. However, the current paper suggests a different perspective. The Abstract and Introduction refer to the significance of mouse models, such as Vk*MYC, to help understand the differences between stable vs. progressive precursor conditions, by overcoming the limitation of the long lead time between monoclonal gammopathy and its transformation to MM in patients. Page 5 also mentions that the non-synonymous mutational burden was similar between de novo samples and human stable precursor conditions; and between transplanted Vk*MYC cells and human MM and progressive precursor conditions.

Do the authors feel, based on their genomic data, that de novo, transplanted and in vitro Vk*MYC cells do not reflect newly diagnosed MM, relapsed/refractory MM and human cell lines, respectively, but rather stable precursors vs. progressive precursors/symptomatic MM vs. relapsed/refractory MM (or maybe human cell lines)?

If they consider that the genomic data give a clear answer, this should be emphasized, as it would be a major conclusion of the paper and key information for the readers.

If their view is that a clear answer is not possible at this point due to technical or conceptual reasons, these should also be described.

2. De novo, transplanted and in vitro Vk*MYC cells have different rates and complexity of genomic events. In some cases, though, the study describes only aggregate numbers for these 3 groups combined, not separately (as also indicated in other comments below).

Please provide whenever possible, especially for key results of this study, the breakdown for each group of Vk*MYC samples. When comparing the frequency of genomic lesions in Vk*MYC cells vs. human MM samples, please provide whenever possible comparisons that are "matched" (please also see comment 9), e.g., if the authors conclude that de novo Vk*MYC is equivalent to non-progressive precursor in human individuals, then these two groups should be compared with each other (rather than comparisons of de novo Vk*MYC with newly diagnosed MM or aggregate data from different phases of the MM trajectory). If there are technical and conceptual reasons why such direct "matched" comparison is not possible, please indicate why.

3. The study should provide more detailed information and a clearer conclusion on whether most Vk*MYC tumors represent the equivalent of "hyperdiploid" MM.

a) The text and current data indicate that a minority of Vk*MYC tumors exhibit immunoglobulin

locus translocations similar to those in patients with non-hyperdiploid MM. But the manuscript does not formally conclude that most Vk*MYC tumors reflect the "hyperdiploid MM" seen in patients. Is there a reason for this? If indeed most Vk*MYC tumors are reflecting the "hyperdiploid" human MM, this is an important conclusion that needs to be highlighted and contrasted with the near complete absence of "hyperdiploid" human MM lines.

b) page 7 states that "As additional similarity between human and Vk*MYC MM, translocations involving the immunoglobulin heavy and light chain loci were detected in eight Vk*MYC MM (15.4%) (Figure 3F)." This phrase appears to refer to the % of all Vk*MYC samples that were examined (not de novo only or transplanted only). If this is the case, please clarify this and provide a breakdown of the % for de novo vs. transplanted vs. in vitro Vk*MYC samples.

c) Page 6, "Similarly to human hyperdiploid MM, large trisomies in the Vk*MYC myeloma tended to be acquired in the same time window in 8/10 (80%) cases where this analysis was possible (Figure 2C; Supplementary Table 8)": please provide more information about if/why it was not possible for trisomies to be evaluated from more tumors in this study. Are the trisomic chromosomes identified in Vk*MYC tumor syntenic to those that are typically trisomic in hyperdiploid MM patients?

4. Several parts of the text contain comparative statements for rates of mutations between different groups of Vk*MYC samples; or in Vk*MYC cells vs. samples from human MM or precursor conditions. Some of these statements are accompanied by statistics, others are not. For example, in page 7, the comparisons of rates for different types of CNV lesions (e.g., large and focal gains of chromosome 1, chromosome 3 amplification or monosomy 5) in de novo Vk*MYC MM vs transplanted vs. in vitro Vk*MYC cells should be supported by statistical analyses. In general, for all such comparative statements, please include statistics when possible. When it is not, please indicate (e.g., in methods or supplemental table/figures) why such comparisons cannot be supported by statistics and tone down accordingly any respective statements about the extent of observed differences (or similarities).

5. Page 6 (based on figures 2C,D) concludes that "the Vk*MYC MM molecular time estimates were significantly higher compared to that observed in human MM, suggesting a later acquisition in disease pathogenesis". Although prior studies of key investigators of this study address this topic, it will be easier for the reader if more information is given on e.g., what does molecular time of 1 vs. 0.5 vs. 0 means here. Are these molecular time analyses correcting for the different lifespan (and potentially different mutation rates?) in human vs. mice? Does it also correct for the fact that the Vk*MYC are already born with the lesion that is responsible for the eventual development of the disease?

6. Molecular time analyses (page 6, Fig2): If de novo Vk*MYC lesions develop in mice that are ~1 year old (which is equivalent to ~58 years in humans, i.e., quite close to the median age of MM diagnosis), does this mean that Vk*MYC mice accumulate the full set of genomic lesions required for MM slower than in humans (even though the mice are already born with the "initiating" lesion of MYC)?

7. Apparently 13 of the 84 unique Vk*MYC samples that were examined exhibited no reversion of the mutation that controls expression of MYC. Any insights into how MYC was activated in those tumors?

8. For figure 2A, please try to present (e.g., in Supplementary Figures) alternative versions that focus not on all the 96 samples, but on the de novo or the transplanted ones.

9. Page 8 and figure 3 (structural variants and complex events): The comparison of SVs in Vk*MYC MM vs. human MM apparently involved WGS data from any type of Vk*MYC cells (de novo, transplanted, growing in vitro) and it is not clear if the human MM was newly diagnosed, MGUS/SMM, relapsed/refractory or a combination of them.

Because this section of the paper makes a direct comparison between the Vk*MYC and the human MM, it is important that the comparison is an "apples to apples" comparison, i.e., the de novo Vk*MYC cells are compared with the stage of MM that the authors consider its equivalent; and that similar comparisons are performed for the transplanted Vk*MYC cells or the cells growing in vitro.

10. Retro-transposition of IAP LTR is an interesting mechanism for potential activation of NF-

kappaB in about 20% of Vk*MYC tumors (presumably all forms of Vk*MYC, not just de novo). Please clarify how frequently retro-transposition of IAP LTR occurs in MM patient samples: if it is indeed rare in human MM but frequent in Vk*MYC, what are the possible causes or implications of this difference?

11. Some statements in the Discussion and other parts of the paper, about the extent of genomic similarity between Vk*MYC and human MM, can be toned down, for example, the statement in the Discussion about “the striking genomic similarity between Vk*MYC and human MM”. The study shows many important similarities, but also some important differences, e.g., molecular time; lower number of structural variants in Vk*MYC compared to human MM; lower frequency of Ig translocations; patterns of mutations in certain genes e.g., PTEN high rates of Retro-transposition of IAP LTR, or different frequencies of APOBEC mutational activity. These differences do not lessen the value of the Vk*MYC model or its pathophysiological similarity with human MM, but overall some aspects of the genomic similarities are not as striking as the text sometimes states or implies. This comment can be easily addressed with changes in different parts of the text, but it is important.

Minor comments

A. Page 5 “To identify driver genes under positive selection, the ratio of non-synonymous to synonymous substitutions (dN/dS) was tested using the dnscv R 13 package, both considering all mutations and restricting to a catalogue of oncogenes generated by combining known MM driver genes and the COSMIC census”.

Were there other genes with high ratio of non-synonymous to synonymous substitutions that were not part of the catalog of MM driver genes and the oncogenes from the COSMIC census?

B. page #5, “Positively selected genes not previously described in MM included H1f2/Hist1h1c, Pten, and Tbsb4x”. This can be rephrased because PTEN is probably not a frequent target for genomic lesions in MM, but this is not the first time that it is implicated in MM biology in at least some fashion.

C. Page #5, “Additional patterns of co-occurrence between mutated driver genes were observed such as between Tent5c and Hif4 ($p < 0.05$ using Fisher’s exact test; Supplemental Figure 2C).” Please check again the spelling of gene symbols in this figure and in the text (Supplementary Table 4 does not list any mutations in a Hif4 gene, it is probably H1f4 gene. Also in the same Supplemental Figure 2C, please check the spelling for Tent5c.

D. A suggestion for Figures 2A, 2B (and potentially several others): can you introduce (e.g., on the margin of the figure) some info which chromosomal locations in human genome are syntenic to the indicated mouse chromosomal regions? This would help readers that are less familiar with mouse genomics, and primarily focused on genomics of human MM.

E. Figure 2A,B: Can you introduce in a supplementary figure a comparison of GISTIC plots (2A) and heat map (2B) from the Vk*MYC data vs. similar plots for the human data?

F. Page 8 (conclusion of section on “Structural variants and complex events”): “Overall, we found significant similarities between the SV landscape of human MM and Vk*MYC myeloma, including complex events such as chromothripsis and translocations involving the immunoglobulin genes”. Please consider rephrasing this statement to indicate that there are also differences (e.g., the lower rate of SVs in the Vk*MYC vs. MM, if indeed this result holds true after analyses matched for the respective stage of myeloma genesis) and also the fact that the immunoglobulin gene translocations are more infrequent than in human MM.

G. In Fig 5D-E, the experiment for ongoing MYC dependence involved a measurement of Ig levels 5 days after the start of tamoxifen treatment. Were any measurements taken at later time points? Did Vk22284 tumors eventually grow out after tamoxifen treatment and the loss of the MYC transgene?

H. Page 12, “Most interestingly, APOBEC mutational activity was detected in 44% of Vk*MYC MM

WGS, a proportion slightly lower compared to that observed in human MM and progressive MM precursor conditions (~80%; Figure 6E,F)". This sentence should be rephrased as it is confusing. For instance, it cannot be clearly deduced from Fig. 6 E or F that the % WGS samples with APOBEC mutational activity was 44% for of Vk*MYC MM or 80% for human MM and progressive precursor conditions. Can the authors explain why a difference of 44% vs 80% can be considered in this case to be only "slight"? Perhaps this needs to be rephrased?

Reviewer #2 (Remarks to the Author):

In „The Vk*MYC Mouse Model recapitulates human multiple myeloma evolution and genomic diversity" Maura and co-workers profiled genetically engineered Vk*MYC mice to reveal if their genomic profiles and evolution pathways correspond to the ones seen in human myeloma. They show several similarities between mice and multiple myeloma, including APOBEC mutational activity and large chromosomal gains, and conclude that these mouse models recapitulate human myeloma evolution and genomic diversity, making them excellent models to decipher the mechanisms underlying disease progression. My comments are included below:

1) I believe that there is a conflict between the introduction and the study design. In the introduction the authors wrote that "the differentiation between progressive and stable precursor conditions is one of the most important unmet clinical needs in the MM community", "makes investigations focusing on early phase of cancer development a significant challenge" and "to overcome this challenge". Hence, according to the introduction the authors were interested in the key genomic events driving (early) myeloma progression. However, in this study they compared de novo mice to transplanted mice and cell cultures of these tumors. According to the introduction de novo mice correspond to treatment requiring human myeloma, rather than to a human precursor condition. Therefore, I suggest to either rewrite the introduction, since the results for the presented comparisons are indeed interesting, or to also compare de novo to "young" Vk*MYC without end organ damage.

2) Some lines, e.g. Vk36040, were interrogated at different stages. Yet, the results are not discussed although some of them show changes between stages, such as a difference in the mutational signatures in Vk36040 (+SBS13).

3) I understand why the authors combined all samples to perform the driver gene analysis. Yet, it seems to be that most if not all of the identified drivers are present in advanced stages of the mouse model, making them candidates as drivers of aggressive disease and proliferation. As described above, the authors seemed to be interested in drivers of early myelomagenesis. Thus, this potential difference also needs to be discussed.

4) The number of variants per Mb per sample did not show a difference between WGS and WES. However, there seems to be a difference between the number of driver mutations in transplanted mice analyzed by WGS or WES as shown in Fig 1A. Is that just by chance or could that also be due to a change in the microbiome occurring in the colony, as assumed for the isotype, or any other reason?

5) Mutations in Tp53 and Dusp2 were enriched in tumors growing in vitro compared to transplanted and de novo combined. Why did the authors combine the latter two? These mutations were also frequent in transplanted mice.

6) Large trisomies tended to be acquired in the same time window. Were these always the same chromosomes or groups of chromosomes as seen in human myeloma? Do these trisomies have a similar impact on gene expression as described for human myeloma? That would be important to conclude that there is similarity between mice and human MM, since groups of trisomies are also seen in other entities, such as CLL.

7) Please add statistics/values to the statement that molecular time estimates in mice were higher compared to what observed in human MM.

8) I don't agree that there is a general acceleration in tumor development in male compared to female mice. According to Fig. 2F there only seems to be a slight delay for female mice between weeks 50 and 60 and afterwards the curves are rather parallel. Is that a subgroup-specific effect?

9) The authors need to be careful that they don't overinterpret their findings. There are similarities but they are not striking. There is pronounced heterogeneity between mice, so they (as a group) are not similar to the 30-40% of human myeloma without primary IgH translocations and presence of MYC translocations (and presence of trisomies). It rather seems to be that some of them are similar, while others completely differ. Furthermore, chromosomal instability is a hallmark of cancer

in general. Finally, as already discussed by the authors, the timing of some important events is opposite in mice and human MM. Therefore, I even recommend changing the title of the manuscript, since the model does not fully recapitulate human myeloma evolution.

10) The authors wrote that MYC activation was a secondary genomic and progression event. I'm not sure if these are the correct terms. The MYC translocation is rather a tumor-initiating somatic event in this model (such as an IgH translocation in human MM) but seems to interact with/ is dependent on germline variants.

11) The authors wrote that the low incidence of RAS mutations and the high frequency of Pten inactivation could be due to a more significant role of the MTORC pathway than MAPK. This is poorly speculative and should be removed. It rather seems to be that there are some significant differences between human MM and the mouse model.

12) In Fig. 2C Vk32829 is shown as an example for the molecular time analysis. Vk32829 is a transplanted tumor. Does that mean that the trisomies were acquired after the transplant? Please indicate the time to de novo and progression after transplant in the figure. Did the authors select this case as representative for a simultaneous gain of trisomies? It seems to be that trisomies were acquired in different time windows.

13) In Fig. 5C the authors show a correlation between proliferation and the level of Ig transcription. Did the authors account for differences in sample purity? And do they also see a correlation if they focus on newly diagnosed MM?

14) Which sample(s) did the authors use as normal control for sequencing?

We thank the Reviewers for their constructive comments. After thorough revision, we are pleased to present an updated version of our manuscript titled: “The Vk*MYC Mouse Model recapitulates human multiple myeloma evolution and genomic diversity”, that at request of reviewer 2 we have retitled: *The heterogeneous genomic landscape of Vk*MYC myeloma highlights shared pathways of plasma cell malignant transformation between mice and humans.*

We thank the reviewers for pointing out the inconsistency between title, introduction and study design. Both reviewers 1 and 2 are correct in their thinking that our study does not address the transformation from MGUS to MM, but rather focuses on the characterization of genomic events that are present in indolent (*de novo*) MM and that drive the progression to aggressive (transplant) and *in vitro* MM, or, in other words, from NDMM to RRMM and eventually MM that grows *in vitro*. In the introduction of the previous version of this manuscript we gave the impression that our data would delve into the genomic evolution of the Vk*MYC model from MGUS to MM. Wild type C57BL/6 mice are the model to study MGUS, because they develop a stable gammopathy with age, that for example we have shown can be anticipated by heterozygous deletion of the *muMIR15a/16-1* cluster. In contrast, Vk*MYC mice are not a model of stable MGUS, because MYC expression inexorably leads to its progression. However, they can be used to model the early phase of progression of a monoclonal gammopathy to SMM and characterize the spontaneously acquired tumor driver mutations and microenvironment changes that lead to MM progression. We have rephrased the abstract and introduction accordingly.

Below, we include a point-by-point response– the Reviewers’ comments are in black, our response is in blue.

=====

REVIEWER #1

This manuscript by Maura et al. is a meaningful contribution and important resource for research in multiple myeloma (MM) and other plasma cell neoplasias. The Vk*MYC model and the different version of its tumor cells exhibit many features that faithfully recapitulate the pathophysiology of MM. Vk*MYC tumor cells (*de novo*, transplanted, or *in vitro* cultured) are extensively used to study MM biology of MM and response to treatments, so this detailed assessment of their genomic

features is important. For this reason, it is especially important that the manuscript addresses more clearly some of its key questions, e.g. which stages in the "life history" of MM are recapitulated by the different versions of Vk*MYC tumor cells (de novo, transplanted, or in vitro cultured); whether the Vk*MYC model in general reflects many genomic subtypes of MM but primarily those cases which lack primary IgH translocations and if, by extension, the Vk*MYC model can be considered a model of "hyperdiploid MM". To their credit, the authors provide data for not only for those genomic features that are similar in Vk*MYC model and human MM, but also for some that appear distinct. Overall, this is an important resource for myeloma research and hopefully addressing the comments below will help enhance it.

We thank the Reviewer for his/her positive comments. In the new version of the manuscript, we have carefully revised the text and clarified the important points and questions raised by the Reviewer. Specifically:

1. We have explicitly stated that the Vk*MYC MM samples investigated in our study, analyzed using Whole Genome Sequencing (WGS) and Whole Exome Sequencing (WES), represent active multiple myeloma (MM) rather than precursor conditions. See above the introductory statement.
2. After careful consideration, we acknowledge that while there are notable similarities, we have refrained from categorizing Vk*MYC MM as hyperdiploid MM. Consequently, the reference to Vk*MYC MM as hyperdiploid MM has been omitted from our discussion. In addition, we have included as **Supplementary Table 21** a table comparing common genomic abnormalities across Vk*MYC and the different subtypes of human MM.
3. It is essential to emphasize that due to its inherent heterogeneity, the presence of multiple genomic drivers that spontaneously emerge over time, and the genomic resemblance to human MM, the Vk*MYC mouse model has emerged as an invaluable platform for future research. This model is poised to address several of the known limitations in human MM research, including the ability to perform longitudinal sampling and overcome the significant time gap between disease initiation and clinical diagnosis.

Main comments:

1. It is important that the paper provides a clearer statement on which parts of the "life history of MM" are represented, based on the current genomic data, by de novo vs. transplanted vs.

in vitro growing Vk*MYC cells. Through the prior work of Drs. Chesi and Bergsagel on the Vk*MYC model, a widely held impression (perhaps incorrect?) in the MM field is that de novo Vk*MYC lesions simulate newly diagnosed MM; transplanted Vk*MYC lesions are reminiscent of relapsed and refractory MM; and Vk*MYC MM cells growing in vitro are the mouse equivalent of human MM lines. However, the current paper suggests a different perspective. The Abstract and Introduction refer to the significance of mouse models, such as Vk*MYC, to help understand the differences between stable vs. progressive precursor conditions, by overcoming the limitation of the long lead time between monoclonal gammopathy and its transformation to MM in patients.

We agree with the reviewer that the Vk*MYC model cannot be used to model a stable monoclonal gammopathy, but rather the MGUS/SMM that will inevitably progress into MM. We never consider *de novo* Vk*MYC mice as having stable MGUS, because their disease is never stable but continuously progressing. However, the *de novo* samples that we chose to analyze in this manuscript are mostly representative of newly diagnosed MM, as opposed to SMM, since mice were selected for high tumor burden which we found associated with anemia and therefore allowed us to identify the genomic drivers of the transition between SMM and NDMM. On the other hand, the transplant tumors clearly represent a more aggressive disease that mimic relapse/refractory MM, and the *vitro* lines share many features with human MM cell lines. We have modified the abstract and the manuscript to make it clear that the transition between *de novo* and transplant Vk*MYC MM mimics the transition from NDMM to RRMM in human.

Page 5 also mentions that the non-synonymous mutational burden was similar between *de novo* samples and human stable precursor conditions; and between transplanted Vk*MYC cells and human MM and progressive precursor conditions. Do the authors feel, based on their genomic data, that *de novo*, transplanted and in vitro Vk*MYC cells do not reflect newly diagnosed MM, relapsed/refractory MM and human cell lines, respectively, but rather stable precursors vs. progressive precursors/symptomatic MM vs. relapsed/refractory MM (or maybe human cell lines)? If they consider that the genomic data give a clear answer, this should be emphasized, as it would be a major conclusion of the paper and key information for the readers. If their view is that a clear answer is not possible at this point due to technical or conceptual reasons, these should also be described.

We are not concerned by the observation that *de novo* Vk*MYC tumors are more similar to stable precursor disease (**Figure 1B**). On the other hand, based on this fact alone we don't believe the *de novo* Vk*MYC models stable precursor disease, considering that their disease may be indolent but is never stable. We speculate that perhaps, similarly to human MM, the disease progression and transformation is mostly promoted by CNV, SV and insertional mutagenesis rather than nonsynonymous single nucleotide variants.

2. De novo, transplanted and in vitro Vk*MYC cells have different rates and complexity of genomic events. In some cases, though, the study describes only aggregate numbers for these 3 groups combined, not separately (as also indicated in other comments below). Please provide whenever possible, especially for key results of this study, the breakdown for each group of Vk*MYC samples.

In response to the Reviewers' insightful comments, we have incorporated new supplementary tables and figures to present the results of rigorous statistical tests that compare various aspects of the genomic landscape in our study (i.e. **Supplementary Tables, 5, 6, 10, 11 and 15 and Supplementary Figure 12**). Specifically, these tables provide a comprehensive analysis of single nucleotide variants (SNVs), copy number variants (CNVs), structural variants (SVs), and mutational signatures across all potential combinations of *de novo*, transplant, and *in vitro* Vk*MYC tumors.

- When comparing the frequency of genomic lesions in Vk*MYC cells vs. human MM samples, please provide whenever possible comparisons that are "matched" (please also see comment 9), e.g., if the authors conclude that *de novo* Vk*MYC is equivalent to non-progressive precursor in human individuals, then these two groups should be compared with each other (rather than comparisons of *de novo* Vk*MYC with newly diagnosed MM or aggregate data from different phases of the MM trajectory). If there are technical and conceptual reasons why such direct "matched" comparison is not possible, please indicate why.

As stated above, we do not believe that *de novo* Vk*MYC models non-progressive precursor disease, but rather NDMM.

We have made the following comparison between Vk*MYC and human MM:

1. In **Figure 1B** and **Supplementary Table 5** we compare non synonymous SNV between different stages of Vk*MYC and human MM. See comments above.
2. In **Figure 5B,C** we compare Ig transcription and proliferation across human and mouse disease stages.
3. In **Figure 6F** we compare APOBEC mutational contribution across different human MM stages and Vk*MYC MM. In **Supplementary Figure 12** we then break down APOBEC signature in different stages of Vk*MYC MM. This distinction is particularly notable, given the marked occurrence of spontaneous APOBEC mutagenesis in the Vk*MYC model, which sharply contrasts with the complete absence of these signatures in stable MGUS and SMM and in other MM mouse models. This contrast underscores the unique characteristics of the Vk*MYC model and its relevance to understanding MM pathogenesis.
4. In **Supplementary Table 21** we compare genomic events across Vk*MYC transplant, for which we have a larger dataset than de novo Vk*MYC MM (only two WGS analyzed) to human NDMM, for which the CoMMpass dataset provides an adequately sized dataset to break down in the individual disease subgroups, that is not available for RRMM.

We did not directly break down and compare the frequency of all the individual structural variation, copy number changes, non synonymous SNV across different stages of human and Vk*MYC since we did not have enough samples for this analysis.

3. The study should provide more detailed information and a clearer conclusion on whether most Vk*MYC tumors represent the equivalent of “hyperdiploid” MM.

We refrain from stating that Vk*MYC models HRD MM, as it does not fit within a particular subtype of human MM and it does not have the typical pattern of recurrent trisomies. Rather, Vk*MYC MM shares features present in the 10% of human MM having MYC translocations and CCND2 expression but lacking primary recurrent Ig translocations. This group is primarily comprised of HRD11- MM, but also contains a fraction of patients that lacks hyperdiploidy and has high frequency of NFkB activating mutations, which has been classified as nHRD2 (de Leval at al. Blood 2022 PMID 36001803). **Supplementary Table 21** summarizes similarities and differences between Vk*MYC and different subtypes of human MM.

a) The text and current data indicate that a minority of Vk*MYC tumors exhibit immunoglobulin locus translocations similar to those in patients with non-hyperdiploid MM. But the manuscript does not formally conclude that most Vk*MYC tumors reflect the “hyperdiploid MM” seen in patients. Is there a reason for this? If indeed most Vk*MYC tumors are reflecting the “hyperdiploid” human MM, this is an important conclusion that needs to be highlighted and contrasted with the near complete absence of “hyperdiploid” human MM lines.

Primary translocations (most commonly dysregulating *CCNDs*, *MAFs*, *NSD2/FGFR3*) in human MM are almost always into the heavy chain switch region, while translocations into light chain loci are predominantly secondary and found in both HRD and non-HRD MM. In Vk*MYC MM we detected Ig translocations in both heavy and light chain loci; and half of the ones in the heavy chain are in the switch region, which could indicate a primary origin (see new **Supplementary Table 16**). However, two of those cause the disruption of the target genes (*Aicda* and *Traf3*), an event never reported in human MM and most likely associated with gene loss of function. We conclude that primary translocations events are less common in Vk*MYC than human MM. See comment above about defining Vk*MYC as a model of HRD MM.

A total of 19 mice with available WGS had large clonal trisomy with molecular time. Among these, 2 had Ig translocations. We can deduce that a subset of mice developed MM through the concurrent acquisition of large chromosomal gains, similarly to what seen in MM pathogenesis. In contrast to MM these events are typically acquired in the later stages of the disease. It's worth noting that in human MM, *MYC* translocations often co-occur with HRD. This leads us to speculate in the discussion that the timing of these two events might be reversed in Vk*MYC myeloma. However, similar to humans MM with *MYC* involvement, not all Vk*MYC cases exhibit the HRD CNV profile. This spontaneous heterogeneity observed in Vk*MYC myelomagenesis is a critical and highly relevant aspect of this study. It underscores the model's strong resemblance to human MM, its ability to capture the inherent heterogeneity seen in the human disease and closely mirrors its behavior in humans. As stated above, even if we observed interesting similarities, following Reviewer #1 and #2 comments we decided to remove the sentence regarding the link between human HRD MM and Vk*MYC MM.

b) Page 7 states that “As additional similarity between human and Vk*MYC MM, translocations involving the immunoglobulin heavy and light chain loci were detected in eight Vk*MYC MM (15.4%) (Figure 3F).” This phrase appears to refer to the % of all Vk*MYC samples that were

examined (not de novo only or transplanted only). If this is the case, please clarify this and provide a breakdown of the % for de novo vs. transplanted vs. in vitro Vk*MYC samples.

Similar to CNV (in **Supplementary Table 10** and **11**) and SNV (in **Supplementary Table 5** and **6**), we have provided in **Supplementary Table 15** a detailed breakdown and statistical comparisons for SV between the different stages of Vk*MYC mice. It's important to note that the number of *de novo* Vk*MYC mice with WGS data was too limited to conduct a comprehensive comparison. Therefore, we focused our analysis on comparing transplanted mice to in vitro models as these were the only groups with a sufficient sample size for statistical testing. The manuscript now reports only the comparison with a $p < 0.05$ using Fisher's exact test and Wilcoxon test for categorical and for linear features, respectively. Overall, we did not observe any significant differences in Ig translocation prevalence among the three stages of our study. Specifically, we found that 1 out of 2 *de novo*, 5 out of 30 transplants, and 2 out of 20 *in vitro* samples exhibited at least one of these events.

c) Page 6, "Similarly to human hyperdiploid MM, large trisomies in the Vk*MYC myeloma tended to be acquired in the same time window in 8/10 (80%) cases where this analysis was possible (Figure 2C; Supplementary Table 8)": please provide more information about if/why it was not possible for trisomies to be evaluated from more tumors in this study. Are the trisomic chromosomes identified in Vk*MYC tumor syntenic to those that are typically trisomic in hyperdiploid MM patients?

Unfortunately, our molecular time workflow can only be applied using WGS data on chromosomal gains larger than 1 Mb with more than 50 clonal mutations. These established criteria are the reason why we cannot estimate the relative time of all the large gains in mice with WGS data. The full list of all 34 tested chromosomes and their molecular time data have been included as **Supplementary Table 12** and **Supplementary Figure 6**. The trisomies in Vk*MYC tumors include portions of those trisomic in human disease as well as larger portions that are not. The comprehensive syntenic map is shown as **Supplementary Figure 4**. Overall, in Vk*MYC MM there is not a consistent pattern of trisomies as the one observed in human.

4. Several parts of the text contain comparative statements for rates of mutations between different groups of Vk*MYC samples; or in Vk*MYC cells vs. samples from human MM or precursor conditions. Some of these statements are accompanied by statistics, others are not.

For example, in page 7, the comparisons of rates for different types of CNV lesions (e.g., large and focal gains of chromosome 1, chromosome 3 amplification or monosomy 5) in de novo Vk*MYC MM vs transplanted vs. in vitro Vk*MYC cells should be supported by statistical analyses. In general, for all such comparative statements, please include statistics when possible. When it is not, please indicate (e.g., in methods or supplemental table/figures) why such comparisons cannot be supported by statistics and tone down accordingly any respective statements about the extent of observed differences (or similarities).

In addition to **Supplementary Tables, 5, 6, 10, 11 and 15**, and **Supplementary Figure 12**, we have updated the text and highlight only comparisons that were statistically significant using Fisher's exact test.

5. Page 6 (based on figures 2C, D) concludes that “the Vk*MYC MM molecular time estimates were significantly higher compared to that observed in human MM, suggesting a later acquisition in disease pathogenesis”. Although prior studies of key investigators of this study address this topic, it will be easier for the reader if more information is given on e.g., what does molecular time of 1 vs. 0.5 vs. 0 means here. Are these molecular time analyses correcting for the different lifespan (and potentially different mutation rates?) in human vs. mice? Does it also correct for the fact that the Vk*MYC are already born with the lesion that is responsible for the eventual development of the disease?

We thank the Reviewer for asking these relevant questions. Molecular time is an indirect and relative measure that allows us to estimate when large chromosomal gains were acquired in the patient (or mouse) life-history. It is determined by calculating the corrected ratio between duplicated and non-duplicated clonal mutations within these chromosomal gains. A molecular time of 0 indicates that the gain was acquired early in the life of the cell, whereas a value of 1 suggests acquisition around the time of sample collection. This is now clarified in the **Figure 2C-D**.

Recent research by the Martincorena Lab (Cagan A et al., Nature 2022; PMID: 35418684) has demonstrated that the SBS1 and SBS5 mutation burden in mice is similar to that in humans, reflecting an accelerated mutation rate. Additionally, other recent studies, such as Riva et al., Nature Genetics 2020 (PMID: 32989322), have shown that the mutational signatures observed in mice exposed to mutagenic agents closely mimic to those observed in humans. Furthermore,

aside from a few specific genomic events (e.g., MAF/MAFB, POLY, MSI, and BRCA), the majority of genomic drivers in tumors do not significantly impact the mutation rate (Gerstung et al., Nature 2020; PMID: 32025013). Considering these factors, we do not anticipate that the life span of mice would significantly affect our molecular time calculations and the comparison with humans. We have included these considerations in the updated version of the manuscript to provide a comprehensive context for our methodology.

6. Molecular time analyses (page 6, Fig2): If de novo Vk*MYC lesions develop in mice that are ~1 year old (which is equivalent to ~58 years in humans, i.e., quite close to the median age of MM diagnosis), does this mean that Vk*MYC mice accumulate the full set of genomic lesions required for MM slower than in humans (even though the mice are already born with the “initiating” lesion of MYC)?

The Reviewer's observation is indeed valid. To clarify, we would use "later" instead of "slower" to accurately convey the intended meaning. Additionally, it's essential to emphasize that this statement specifically pertains to large chromosomal gains, which aligns with what is observed in human MM. In human patients, the timing of canonical IGH translocations and HRD events is significantly different. When these two events co-occur, the IGH translocation event typically occurs earlier in the disease progression. Conversely, if HRD is the initial event (i.e., HRD without IGH translocations), the molecular time is considered early. Therefore, the timing of large chromosomal gains is contingent on which event occurs first, mirroring the dynamics observed in human MM. This insight is supported by data indicating that the molecular time of patients with

both IGH translocations and HRD is later compared to those with HRD alone, which aligns with the unpublished **Rebuttal Figure 1** and Maura et al. ASH 2023 (<https://doi.org/10.1182/blood-2023-182267>).

7. Apparently 13 of the 84 unique Vk*MYC samples that were examined exhibited no reversion of the mutation that controls expression of MYC. Any insights into how MYC was activated in those tumors?

We have determined that all Vk*MYC MM tumors, including the ones lacking reversion of the stop codon, express MycS, a N-terminus truncated MYC isoform previously observed in several cell lines and originating from conserved ATG translational initiation codons downstream of the canonical initiation site (Spotts 1997 PMID:9032273; Oster 2003 PMID:12673205; Benassayag 2005 Molecular and Cellular Biology,25:22; Cowling, 2008 PMID:17704800), (**Rebuttal Figure 2**). Interestingly, MycS was found to be able to induce transformation in Rat1a cells, but weaker than full-length MYC in inducing cell proliferation in vitro. Consistently, we have generated Vk*MYC_S transgenic mice constitutively expressing MycS only (unable to express full-length MYC) and found that they develop a slowly progressing monoclonal gammopathy that never progressed to aggressive and transplantable MM. We have also generated another transgenic mouse line (Vk*MYC_FL) expressing the full-length MYC only and unable to express MYCS. These mice develop full blown MM at a slightly lower incidence, and generated transplantable lines. We concluded that MYCS is sufficient to induce plasma cell proliferation, but not to drive malignant transformation. Consistently, reversion of the stop codon is selected in the majority of Vk*MYC MM. Although interesting, we believe these data are outside the scope of this manuscript and we plan to report them separately.

8. For figure 2A, please try to present (e.g., in Supplementary Figures) alternative versions that focus not on all the 96 samples, but on the de novo or the transplanted ones.

We acknowledge the significance of comparing the CNV landscape across the different stages. However, we have opted not to run GISTIC analysis for each subgroup. GISTIC tends to perform better with larger sample sizes and a more robust background model. Consequently, we have retained **Figure 2A** as it is. Nevertheless, as previously mentioned, we have included **Supplemental Tables 10** and **11** which offer a statistical comparison of recurrent monosomies and trisomies, focal and large CNVs detected by GISTIC for the three different stages (i.e, de novo, transplant, and in vitro Vk*MYC tumors). This data, in conjunction with **Figure 2B**, should collectively provide a comprehensive overview of the CNV distribution across the different stages, despite not running GISTIC for each subgroup.

9. Page 8 and figure 3 (structural variants and complex events): The comparison of SVs in Vk*MYC MM vs. human MM apparently involved WGS data from any type of Vk*MYC cells (de novo, transplanted, growing in vitro) and it is not clear if the human MM was newly diagnosed, MGUS/SMM, relapsed/refractory or a combination of them. Because this section of the paper

makes a direct comparison between the Vk*MYC and the human MM, it is important that the comparison is an “apples to apples” comparison, i.e., the de novo Vk*MYC cells are compared with the stage of MM that the authors consider its equivalent; and that similar comparisons are performed for the transplanted Vk*MYC cells or the cells growing in vitro.

We agree with the Reviewer. The comparison was based only on NDMM enrolled in CoMMpass (Rustad et al. BCD 2020; PMID: 33392515). This is now clarified in the manuscript. We analyzed 50 transplant and *in vitro*, but only two *de novo* MM for SV. We agree with the reviewer that the ideal human comparison would be a dataset of RRMM and HMCL, however the only large well-characterized dataset for SV in human MM is CoMMpass, from NDMM. As we are finding a lower incidence compared to NDMM, we can conclude that we will also have lower incidence compared to RRMM or HMCL.

As stated above, we also included **Supplementary Table 15** where each SV event or involved driver gene is compared between in vitro and transplant Vk*MYC.

10. Retro-transposition of IAP LTR is an interesting mechanism for potential activation of NF-kappaB in about 20% of Vk*MYC tumors (presumably all forms of Vk*MYC, not just de novo). Please clarify how frequently retro-transposition of IAP LTR occurs in MM patient samples: if it is indeed rare in human MM but frequent in Vk*MYC, what are the possible causes or implications of this difference?

We added this sentence to the discussion “*Retrotransposition is much more common in mouse than human likely because mice have a larger group of active endogenous retroviral elements, including IAP. In contrast human have only a few active transposable elements (TE)s, mainly LINEs and SINEs. Genome wide analysis of human TE has been conducted in multiple cancer, but in contrast with solid tumors, hematological malignancies rarely present these genomic events.*³⁷”. In the cited manuscript which included Drs. Maura, Bolli and Campbell as authors, MM was analyzed but the result was not published since the samples were not a part of the PCAWG). The significance is that although the mice may utilize unique mechanisms, the end result is to dysregulate pathways important for disease progression that are shared with human disease.

11. Some statements in the Discussion and other parts of the paper, about the extent of genomic similarity between Vk*MYC and human MM, can be toned down, for example, the statement in the Discussion about “the striking genomic similarity between Vk*MYC and human MM”. The

study shows many important similarities, but also some important differences, e.g., molecular time; lower number of structural variants in Vk*MYC compared to human MM; lower frequency of Ig translocations; patterns of mutations in certain genes e.g., PTEN high rates of Retro-transposition of IAP LTR, or different frequencies of APOBEC mutational activity. These differences do not lessen the value of the Vk*MYC model or its pathophysiological similarity with human MM, but overall some aspects of the genomic similarities are not as striking as the text sometimes states or implies. This comment can be easily addressed with changes in different parts of the text, but it is important.

Following Reviewer comments, we carefully reviewed the manuscript and toned-down statements not fully supported by the data.

Minor comments

- A. Page 5 “To identify driver genes under positive selection, the ratio of non-synonymous to synonymous substitutions (dN/dS) was tested using the dndscv R 13 package, both considering all mutations and restricting to a catalogue of oncogenes generated by combining known MM driver genes and the COSMIC census”. Were there other genes with high ratio of non-synonymous to synonymous substitutions that were not part of the catalog of MM driver genes and the oncogenes from the COSMIC census?

There were no identified genes under positive selection from our dN/dS analysis that were not a part of the MM driver genes and the COSMIC consensus. Below (**Rebuttal Figure 3**) is a comparison of the results of the dN/dS analysis with and without restriction of known oncogenes.

Supplemental Figure 2B has been updated to indicate which genes became significant after

restricting dN/dS analysis with known oncogenes. This is expected considering that the identification of new oncodriver (e.g., not included in COSMIC) using dNdS would be possible if hundreds of samples were available.

- B. page #5, “Positively selected genes not previously described in MM included H1f2/Hist1h1c, Pten, and Tbsb4x”. This can be rephrased because PTEN is probably not a frequent target for genomic lesions in MM, but this is not the first time that it is implicated in MM biology in at least some fashion.

The statement has been revised to: “*Positively selected genes currently not included in the catalogue of recurrent MM drivers included H1f2/Hist1h1c, Pten, and Tbsb4x.*”

- C. Page #5, “Additional patterns of co-occurrence between mutated driver genes were observed such as between Tent5c and Hif4 ($p < 0.05$ using Fisher’s exact test; Supplemental Figure 2C).” Please check again the spelling of gene symbols in this figure and in the text (Supplementary Table 4 does not list any mutations in a Hif4 gene, it is probably H1f4 gene. Also, in the same Supplemental Figure 2C, please check the spelling for Tent5c.

We have corrected the Hif4 spelling mistake in the text. Since *Tent5c* and *Fam46c* as well as *H1f4* and *Hist1h1e* are used interchangeably throughout the historical scientific literature, we have referred to them within the revised text as follows: *Tent5c/Fam46c* and *H1f4/Hist1h1e*. Within the figures and tables, we have chosen to use the currently accepted gene symbol: *Tent5c* and *H1f4*.

- D. A suggestion for Figures 2A, 2B (and potentially several others): can you introduce (e.g., on the margin of the figure) some info which chromosomal locations in human genome are syntenic to the indicated mouse chromosomal regions? This would help readers that are less familiar with mouse genomics, and primarily focused on genomics of human MM.

We have provided a new **Supplemental Figure 4** showing all chromosomal regions with CNV in Vk*MYC mice that are syntenic to humans. In addition, in **Supplementary Table 8** we have reported all the genes within each GISTIC peak. In figure 2A we labeled the peaks with MM relevant genes.

- E. Figure 2A,B: Can you introduce in a supplementary figure a comparison of GISTIC plots (2A) and heat map (2B) from the Vk*MYC data vs. similar plots for the human data?

Following reviewer comments, we have introduced a **Supplementary Figure 5**, analogous to figure 2B, showing a CNV heat map for human MM.

- F. Page 8 (conclusion of section on “Structural variants and complex events”): “Overall, we found significant similarities between the SV landscape of human MM and Vk*MYC myeloma, including complex events such as chromothripsis and translocations involving the immunoglobulin genes”. Please consider rephrasing this statement to indicate that there are also differences (e.g., the lower rate of SVs in the Vk*MYC vs. MM, if indeed this result holds true after analyses matched for the respective stage of myeloma genesis) and also the fact that the immunoglobulin gene translocations are more infrequent than in human MM.

Following reviewers comments we added the sentence: *“In contrast to human MM, Vk*MYC tumors lack recurrent primary IgH translocations such as Ccnd or Maf gene families or Nsd2/Fgfr3. Overall, while variations exist in the prevalence of specific complex events and the participation of distinct oncodrivers, the Vk*MYC SV landscape exhibits intriguing similarities with human MM. Notably, these shared features include occurrences of chromothripsis and translocations involving the immunoglobulin genes, underscoring the relevance of the VkMYC model as a valuable tool for studying MM pathogenesis”.*

- G. In Fig 5D-E, the experiment for ongoing MYC dependence involved a measurement of Ig levels 5 days after the start of tamoxifen treatment. Were any measurements taken at later time points? Did Vk22284 tumors eventually grow out after tamoxifen treatment and the loss of the MYC transgene?

Figure 5E shows the M-spike levels measured four weeks (not 5 days) after initiation of tamoxifen treatment. We have now included in **Supplementary Figure 8D** the complete analysis of M-spikes levels over time, showing that Vk22284 tumor bearing mice started to relapse 6-7 weeks after TAM treatment initiation. We repeated TAM treatment for five additional days without affecting tumor growth. Consistently, competitive PCR on tumor DNA harvested at necropsy from two of the Vk22284 relapsing mice detected the presence of the unfloxed Vk*MYC allele (**Supplementary Figure 8E**), which is likely driving tumor growth. We concluded that Vk22284

tumors are dependent on sustained MYC expression, but tumor cells with incomplete floxing of the Vk*MYC allele are eventually selected over time and drive relapse. We modified the text to include the sentence” “Tumors eventually relapsed despite retreatment with TAM. Molecular analysis showed that all contained unfloxed VkMYC allele, highlighting the selection of MYC expression for tumor growth (**Supplementary Figure 8D,E**)”.

- H. Page 12, “Most interestingly, APOBEC mutational activity was detected in 44% of Vk*MYC MM WGS, a proportion slightly lower compared to that observed in human MM and progressive MM precursor conditions (~80%; Figure 6E,F)”. This sentence should be rephrased as it is confusing. For instance, it cannot be clearly deduced from Fig.6 E or F that the % WGS samples with APOBEC mutational activity was 44% for of Vk*MYC MM or 80% for human MM and progressive precursor conditions. Can the authors explain why a difference of 44% vs 80% can be considered in this case to be only “slight”? Perhaps this needs to be rephrased?

The fraction of Vk*MYC mice with APOBEC mutational signature is shown in **Figure 6A**. A similar plot for human MM has already been published (PMID: 32317634). **Figure 6F** instead reports the proportion of mutational signature due to APOBEC across Vk*MYC and human MM and precursor conditions, which is similar across human MM and Vk*MYC. This sentence has been rephrased, and now reads: “*Most interestingly, APOBEC mutational activity was detected in 44% of Vk*MYC MM by WGS (**Figure 6A**) and, when present its relative contribution was similar to that of human MM (**Figure 6F**). Although this proportion is lower in comparison to the 80% we have previously reported for human MM and its progressive precursor conditions, the significant discovery that Vk*MYC MM spontaneously develops APOBEC mutagenesis is a critical finding, considering that no other mouse model of cancer has been reported to spontaneously acquire an APOBEC mutational signature*”.

REVIEWER #2

In “The Vk*MYC Mouse Model recapitulates human multiple myeloma evolution and genomic diversity” Maura and co-workers profiled genetically engineered Vk*MYC mice to reveal if their genomic profiles and evolution pathways correspond to the ones seen in human myeloma. They show several similarities between mice and multiple myeloma, including APOBEC mutational activity and large chromosomal gains, and conclude that these mouse models recapitulate human

myeloma evolution and genomic diversity, making them excellent models to decipher the mechanisms underlying disease progression. My comments are included below:

- 1) I believe that there is a conflict between the introduction and the study design. In the introduction the authors wrote that “the differentiation between progressive and stable precursor conditions is one of the most important unmet clinical needs in the MM community”, “makes investigations focusing on early phase of cancer development a significant challenge” and “to overcome this challenge”. Hence, according to the introduction the authors were interested in the key genomic events driving (early) myeloma progression. However, in this study they compared de novo mice to transplanted mice and cell cultures of these tumors. According to the introduction de novo mice correspond to treatment requiring human myeloma, rather than to a human precursor condition. Therefore, I suggest to either rewrite the introduction, since the results for the presented comparisons are indeed interesting, or to also compare de novo to “young” Vk*MYC without end organ damage.

We agree with the reviewer and have modified the text of the manuscript accordingly, see our introductory statement of this rebuttal.

Regarding the comparison of young to de novo mice, we completely agree that it will be informative and we are planning to do it in future studies.

- 2) Some lines, e.g. Vk36040, were interrogated at different stages. Yet, the results are not discussed although some of them show changes between stages, such as a difference in the mutational signatures in Vk36040 (+SBS13).

Following Reviewer #1 and #2 comments, we have included multiple new Supplementary Tables where we compared the prevalence and distribution of different somatic events. To explore the genomic evolution in Vk*MYC mice with 2 samples collected at different stages, we have also created a new **Supplementary Figure 6** where we report the clonal and subclonal evolution from one stage to another. To answer to the Reviewer comment, we investigated APOBEC mutational contribution in VK*MYC mice with 2 samples collected at different stages with available either WGS or WES. Overall, we observed APOBEC mutational activity in both stages in 4/5 cases. In Vk36040, as pointed out by the Reviewer was the only case where APOBEC was found in the

transplantable mouse and not in the *de novo*. Based on the concordance of all the other paired samples, we believe that APOBEC mutagenesis might be likely already active but undetectable at the *de novo* stage. However, this cannot be established with confidence with current bulk sequencing data. These considerations have been included in the updated version of the manuscript.

- 3) I understand why the authors combined all samples to perform the driver gene analysis. Yet, it seems to be that most if not all of the identified drivers are present in advanced stages of the mouse model, making them candidates as drivers of aggressive disease and proliferation. As described above, the authors seemed to be interested in drivers of early myelomagenesis. Thus, this potential difference also needs to be discussed.

In response to the valuable feedback from Reviewers #1 and #2, we have made comprehensive language revisions in the updated manuscript. Additionally, wherever possible, we have included subset analyses in the supplementary materials.

We acknowledge the notable observation that the enrichment of distinct mutations in the *in vitro* Vk*MYC model may suggest a potentially more aggressive disease, mirroring human findings. In human cases, all cell lines exhibit mutations in *TP53*, while less than 5% of NDMM cases do. These findings collectively indicate that Vk*MYC MM is primarily influenced by CNV and SV in its early stages, rather than SNV/indels. This parallel with human MM is now a part of the updated discussion.

- 4) The number of variants per Mb per sample did not show a difference between WGS and WES. However, there seems to be a difference between the number of driver mutations in transplanted mice analyzed by WGS or WES as shown in Fig 1A. Is that just by chance or could that also be due to a change in the microbiome occurring in the colony, as assumed for the isotype, or any other reason?

We observed a statistically significant difference between the number of mutations per Mb when comparing WGS and WES in transplant samples (Wilcoxon P value = 0.014) but not *de novo* (Wilcoxon P value = 0.933) or *in vitro* (Wilcoxon P value = 0.600) samples. Since this difference was only observed in one of three subgroups, our interpretation of these data is that this difference was due to random chance and not related to the choice of sequencing technology.

- 5) Mutations in *Tp53* and *Dusp2* were enriched in tumors growing *in vitro* compared to transplanted and *de novo* combined. Why did the authors combine the latter two? These mutations were also frequent in transplanted mice.

Transplant and *de novo* tumors were combined to differentiate *in vivo* (i.e. immunocompetent system) from *in vitro*. Furthermore, this approach also allows to improve statistical power. In the revised manuscript, we have provided the results of statistical tests for all possible combinations of *de novo*, transplant, and *in vitro* *Vk*MYC* cells (**Supplemental Table 6**). In this analysis, *Trp53* mutations were significantly enriched among *in vitro* tumors compared to both transplant (Fisher P value = 0.001) and *de novo* tumors (Fisher P value = 0.006). In contrast, only *Dusp2* mutations were significantly enriched among *in vitro* tumors compared to *de novo* (Fisher P value = 0.022) but not transplant tumor (Fisher P value = 0.406). Given the reduced statistical power of this subgroup analysis, it's challenging to interpret whether this difference is biologically relevant or not. These findings have been discussed in the updated version of the manuscript.

- 6) Large trisomies tended to be acquired in the same time window. Were these always the same chromosomes or groups of chromosomes as seen in human myeloma? Do these trisomies have a similar impact on gene expression as described for human myeloma? That would be important to conclude that there is similarity between mice and human MM, since groups of trisomies are also seen in other entities, such as CLL.

We did not observe in *Vk*MYC* MM the same trisomic combinations typical of human MM. Unfortunately, our molecular time workflow could only be applied to CNV segments larger than 1 Mb with more than 50 clonal mutations, resulting in a limited number of large gains being tested. We have included a list of all 34 CNV segments from the 19 *Vk*MYC* mice with available whole-genome sequencing in **Supplemental Table 12**.

It is essential to note that a syntenic comparison between humans and mice may be challenging, as these events can encompass up to 50% of the entire genome. We concur with the Reviewer that multiple trisomy can be observed in other B-cell tumors, but the specific combination of MYC and multiple trisomy is a characteristic genomic pattern primarily observed in multiple myeloma.

Regarding the impact of HRD gene expression in humans, it is worth noting that trisomy can induce a cumulative copy number effect. However, it is not a predominant feature, and the historical impact of HRD gene expression was typically the result of supervised models against IGH translocations.

7) Please add statistics/values to the statement that molecular time estimates in mice were higher compared to what observed in human MM.

This has been clarified. We also provided more details about this analysis as requested by Reviewer #1.

8) I don't agree that there is a general acceleration in tumor development in male compared to female mice. According to Fig. 2F there only seems to be a slight delay for female mice between weeks 50 and 60 and afterwards the curves are rather parallel. Is that a subgroup-specific effect?

In human SMM, as recently reported from the Icelandic group, there is an increased prevalence of MM in males versus females, with a difference at the median age of presentation of 60% versus 40% that steadily increases with age, suggesting disease acceleration. We do not see this in the Vk*MYC mouse and agree with the reviewer that "acceleration" is the wrong term. We revised the manuscript to say: "*analysis of M-spike over time identified that males had a highly statistically significant earlier onset of monoclonal gammopathy, with a difference in prevalence at 75 weeks of age of 55% versus 45% (Figure 2E). This is in line with the difference in the relative prevalence in male versus female of 56% to 44% of MGUS in those over 50 years old in Olmsted county (3.7% vs 2.9%), and 63% to 37% of SMM in Iceland (0.67% vs 0.39%)*" (PMID:16571879) and PMID 36747117)

9) The authors need to be careful that they don't overinterpret their findings. There are similarities but they are not striking. There is pronounced heterogeneity between mice, so they (as a group) are not similar to the 30-40% of human myeloma without primary IgH translocations and presence of MYC translocations (and presence of trisomies). It rather seems to be that some of them are similar, while others completely differ. Furthermore, chromosomal instability is a hallmark of cancer in general. Finally, as already discussed by the authors, the timing of some important events is opposite in mice and human MM. Therefore, I even recommend changing the title of the manuscript, since the model does not fully recapitulate human myeloma evolution.

As suggested by Reviewer #1, we acknowledge the need to temper some of our statements, and since we agree that we are not studying evolution in this manuscript, we have modified the title accordingly, which now reads: *“The heterogeneous genomic landscape of Vk*MYC myeloma highlights shared pathways of plasma cell malignant transformation between mice and humans”*.

10) The authors wrote that MYC activation was a secondary genomic and progression event. I'm not sure if these are the correct terms. The MYC translocation is rather a tumor-initiating somatic event in this model (such as an IgH translocation in human MM) but seems to interact with/ is dependent on germline variants.

We have revised the discussion to remove speculation about MYC being a secondary event. The statement now reads: “MYC activation in a permissive genetic background provides the conditions for a transition from a stable to a progressing monoclonal gammopathy”.

11) The authors wrote that the low incidence of RAS mutations and the high frequency of Pten inactivation could be due to a more significant role of the MTORC pathway than MAPK. This is poorly speculative and should be removed. It rather seems to be that there are some significant differences between human MM and the mouse model.

*We agree with the reviewer that our statement was speculative, and changed to read: “Another striking difference between Vk*MYC and human MM is the higher prevalence of mutations of Dusp2 (a negative regulator of Stat3 pathway PMID 26479789), and Pten (a negative regulator of mTORC pathway) and low incidence of mutations of Ras (an activator of MAPK and mTORC1³⁹ pathways) in the mouse”.*

12) In Fig. 2C Vk32829 is shown as an example for the molecular time analysis. Vk32829 is a transplanted tumor. Does that mean that the trisomies were acquired after the transplant? Please indicate the time to de novo and progression after transplant in the figure. Did the authors select this case as representative for a simultaneous gain of trisomies? It seems to be that trisomies were acquired in different time windows.

The Reviewer raised an important consideration. Unfortunately, we cannot provide any estimation of when the gains were acquired in respect to the transplant because the molecular time reflects

a relative estimation with a grade of uncertainty (i.e. confidence of interval). We can define if the gains were acquired early or late, but not the exact moment.

Following Reviewer #1 and #2 comments we included all the molecular time plots and data in **Supplementary Table 13** and **Supplementary Figure 7**.

13) In Fig. 5C the authors show a correlation between proliferation and the level of Ig transcription. Did the authors account for differences in sample purity? And do they also see a correlation if they focus on newly diagnosed MM?

We do not have concerns about sample purity. We can precisely account for the purity of the Vk*MYC tumors, generally >85%, as estimated by flow cytometry, determination of the biallelic deletion of immunoglobulin loci and VAF distribution on diploid regions. We specifically selected for our analysis a human dataset in which samples have been sorted by flow cytometry by Dr. Paiva, distinguishing clonal from non clonal plasma cells, and we confirmed the purity by RNAseq analysis, comparing kappa to lambda transcription. We agree with the reviewer that there is no correlation within each individual human MM subgroup, however the correlation holds true across different stages of MM disease.

14) Which sample(s) did the authors use as normal control for sequencing?

As normal match we used three independent WGS generated from tail DNA obtained from three Vk*MYC mice of different genotype, to account for potential substrain specific germline differences. This is now highlighted in method section and in **Supplementary Table 2**.

REVIEWER COMMENTS

Reviewer #1 (Remarks to the Author):

Overall changes to the manuscript: The new title is appropriate and reflects well the content of the study. Most comments have been addressed satisfactorily. Some additional points that may require clarification are mentioned below.

Major comments

Comment 1: this was addressed by the authors.

Comment 2: The initial submission included several statements about results of direct comparisons between the frequency of mutations, SVs or CNVs in individual stages of Vk*myc compared with human MM. In some cases, statistical assessments of those differences were performed, while in others that information was not clearly visible in the paper. This led me to think that the authors intended to make those direct comparisons of genomics features of human vs. mouse MM, to better reveal the biological relevance of the Vk*myc model.

In their response to this comment, but also in some additional ones (eg comment 4), the authors do highlight the results of statistical tests between comparisons of the different stages of the VkMyc model (eg in Supl Table 6, 10, 11 and in Suppl Fig. 12). However, there responses do not give complete emphasis to the matched comparisons between the individual stages of VkMyc model and the respective group of human samples that they best resemble pathophysiologically. Although Supl Table 5 include such mouse-to-human comparisons for mutations, Supl Table 6, 10, 11 and in Suppl Fig. 12 do not address this information. If this information is in other areas of the paper, it would be great to indicate it. If the comparison is not feasible because the number of samples does not provide sufficient statistically significant power, that should be indicated (some calculation of statistical power, if feasible, would be ideal).

Comment 3: satisfactory response.

Comment 4: see comment 2.

Comment 5-6: acceptable responses

Comment 7: The response of the authors on the mechanism of MYC activation in tumors without reversion of the Vk mutation is really intriguing. Readers of this paper may ask the same question as I did in my comments. Would you consider adding in the text some statement indicating that this mechanism is being examined by ongoing studies that will be reported separately from this manuscript? This is not absolutely required, but a suggestion so that readers are not confused.

Comments 8-11: acceptable responses.

Minor comments

Responses for minor comments A-D, F-H are acceptable.

Comment E recommended a comparison of GISTIC plots (2A) and heatmap (2B) with Vk*myc vs human. The revision includes a Suppl. Fig. 5 with heatmap for mouse data. The point of this comment E was to have the data side by side for human vs Vk*myc datasets. Apparently, the authors felt that since the other data are in different parts of the paper, the visual comparison is not necessary to be direct. I don't have an issue with this, if the authors prefer it that way.

Reviewer #2 (Remarks to the Author):

The authors have addressed most of my comments. I only have only one minor comment. The abstract still gives the impression that the authors used the Vk*MYC mouse model to study the mechanisms underlying progression from precursor stages to multiple myeloma:

"Clinical efforts designed to deconvolute such mechanisms are challenged by the long lead time between monoclonal gammopathy and its transformation to MM. MM mouse models represent an opportunity to overcome this temporal limitation." Please revise the abstract accordingly.

We thank the reviewers for their comments which we have addressed as follows:

Reviewer 1

Comment 2 asks that we correlate different stages of murine myeloma to stages of human myeloma. We have done our best, and I think the most informative figures compare mutation rates, proliferation, and immunoglobulin transcription. In response, we have also added NFkB index to provide an additional factor for comparison in a new Supplementary Figure 8 (reproduced below). We have added the following to the text:

*We also observed a similar increase in NFkB index associated with the progression of both human and mouse MM (**Supplementary Figure 8**)*

These are powerful comparisons because of the large number of samples involved. In contrast comparisons of individual mutations or copy number changes are not generally feasible because of their low rates individually. We have the following responses to concerns raised about the individual tables:

ST6 reports NS-SNV. Two that are significantly increased in the *in vitro* are also significantly increased in HMCL compared to NDMM: TP53, 57% (35/61) vs 2% and DUSP2, 10% (6/61) vs 5%. We have added this information to the text:

This is in line with results in human MM with TP53 mutations in 57% of cell lines vs 2% of NDMM, and DUSP2 mutations in 10% of cell lines versus 5% NDMM based on an WES analysis of 69 human myeloma cell lines (<https://www.keatslab.org>) and the CoMMpass dataset.

ST10 reports whole chromosome arm gains and losses. It is not possible to meaningfully compare these regions to the syntenic human regions as so many different regions are involved, as diagrammed in SF4.

ST11 reports focal CNV. Two show significantly increased frequency in the *in vitro*, 3qF2.1 gain (Mcl1 on 1q21 in humans) and 13qC1 loss (Fam172a on 5q15 in humans). The 5q15 region is not affected by deletion in human MM. Amplification of 1q21 is differentially present in the six different human MM subgroups as summarized in ST21. It is not meaningful to compare the overall incidence of 1q in NDMM and HMCL given the markedly different distribution of primary molecular subtypes between these groups. More relevant is that it is a well-accepted event associated with tumor progression. We have added the following to the text:

Although deletions of the FAM172a region (5q15) do not occur in human MM, amplifications of MCL1 (1q21) are common and have been reported to be acquired with tumor progression (Schmidt, T. M., Fonseca, R., & Usmani, S. Z. (2021). Chromosome 1q21 abnormalities in multiple myeloma. Blood cancer journal, 11(4), 83. <https://doi.org/10.1038/s41408-021-00474-8>).

SF12 shows mutational signatures. There were no significant differences across the mouse stages, obviating the need for a comparison to human stages.

Comment 7. As requested, we have added the following sentence:

"We are actively investigating the mechanism of transformation in the cases without reversion and will report the results separately."

Comment E would prefer a side-by-side comparison of the copy number changes in mouse (Fig 2B) and human (Suppl Fig 5) but defers to our preference. We find they are hard to directly compare and have decided to leave them unchanged.

Reviewer 2

The abstract still gives the impression that the authors used the Vk*MYC mouse model to study the mechanisms underlying progression from precursor stages to multiple myeloma

As suggested, we have removed the following offending sentences from the abstract:

Despite advancements in profiling multiple myeloma (MM), there is limited information on mechanisms underlying disease progression. Clinical efforts designed to deconvolute such mechanisms are challenged by the long lead time between monoclonal gammopathy and its transformation to MM. MM mouse models represent an opportunity to overcome this temporal limitation.

And replace it with: "Multiple myeloma is a heterogeneous disease characterized by frequent MYC translocations. Sporadic MYC activation in the germinal center of genetically engineered Vk*MYC mice is sufficient to induce plasma cell tumors in which a variety of secondary mutations are spontaneously acquired and selected over time."

To adhere to editorial policies, we have shortened the title and the abstract

Supplemental Figure 8. NFkB index by stage in murine and human MM

A For murine MM NFkB Index increases from *de novo* to transplant to *in vitro* (Jonckheere-Terpstra test of ordered differences yields a p-value < 0.05).

B For human MM NFkB Index increases from newly diagnosed to relapse (two-sided t-test p-value < 1e-13).